# Estimating effects of serum vitamin B12 levels on psychiatric disorders and cognitive impairment: a Mendelian randomization study
Tianyuan Lu [1,2,3,4,5,6] ✉ & Andrew D. Paterson [7,8,9] ✉

## Abstract

**Background** Vitamin B12 deficiency can lead to pernicious anemia and has been associated with various neuropsychiatric diseases and cognitive decline. However, it is unclear whether increasing serum vitamin B12 levels can help to prevent the onset of psychiatric disorders and cognitive impairment in the general population.

**Methods** Leveraging large-scale genome-wide association studies (GWASs), we conducted Mendelian randomization (MR) and sensitivity analyses to estimate the potential effects of serum vitamin B12 levels on eight psychiatric disorders, educational attainment and cognitive performance. We conducted additional MR analyses utilizing within-sibship studies to mitigate potential residual confounding effects.

**Results** As a positive control, we confirm that a one standard deviation increase in genetically increased vitamin B12 levels is strongly protective against pernicious anemia (odds ratio, OR = 0.24; 95% CI: 0.15–0.40; $p$-value = 2.1×10$^{-8}$). In contrast, MR estimates of vitamin B12 effects on all eight psychiatric disorders, educational attainment and cognitive performance largely overlap with the null. For example, a one standard deviation increase in genetically predicted vitamin B12 levels is associated with an OR of 1.02 for depression (95% CI: 1.00 – 1.04; $p$-value = 0.11), a 0.0077 standard deviation increase in educational attainment (95% CI: −0.010 – 0.025; $p$-value = 0.39) and a 0.013 standard deviation increase in cognitive performance (95% CI: −0.0088 – 0.035; $p$-value = 0.24). No significant associations are identified in sensitivity analyses excluding pleiotropic genetic instruments or MR analyses based on within-sibship studies.

**Conclusions** Our findings suggest that increasing overall vitamin B12 levels may not meaningfully protect against the investigated psychiatric disorders or cognitive impairment in the general population.

## Plain Language Summary

Low vitamin B12 levels are linked to pernicious anemia and have been associated with several psychiatric and cognitive conditions. However, it is unclear whether increasing B12 levels through supplementation can help to prevent these outcomes. Using genetic data from large studies, we found strong evidence that higher B12 levels protect against pernicious anemia, but no evidence that they reduce the risk of psychiatric disorders or cognitive impairment. Our findings suggest that increasing vitamin B12 levels is unlikely to meaningfully improve mental health or cognitive function in the general population.

Vitamin B12, also known as cobalamin, is an important water-soluble vitamin[1]. Together with folate, vitamin B12 has multiple essential roles in various biological processes, such as DNA synthesis and methylation, maturation of erythrocytes, as well as fatty acid and amino acid metabolism[2][-4]. Vitamin B12 also contributes to the maintenance of the nervous system via the synthesis of myelin and neuronal regeneration[5,6]. For humans, vitamin B12 is obtained exclusively from dietary intake[1].

Vitamin B12 deficiency affects a significant proportion of the global population, including both females and males and across all age groups[7,8]. Vitamin B12 deficiency, defined as serum vitamin B12 levels <148 pmol per litre, has been estimated to affect 2.9% of the adult population in the United States[9]. Untreated vitamin B12 deficiency can lead to pernicious anemia and can result in neurological manifestations, such as peripheral neuropathy, ataxia, various psychiatric disorders, as well as cognitive

impairment[7,8,10]. Among individuals with depression or schizophrenia, randomized controlled trials have shown that folate plus vitamin B12 supplementation may preserve cognitive functions and delay the disease progression[11–13]. However, randomized controlled trials have not elucidated whether increasing vitamin B12 levels through supplementation can help to prevent the onset of psychiatric disorders and cognitive impairment in the general population[14], where the majority do not have clinical vitamin B12 deficiency. Although a longitudinal study indicated that vitamin B12 levels are negatively associated with incident depressive symptoms in older adults[15], these observational associations may be subject to confounding factors that are difficult to fully account for, such as socioeconomic status, lifestyle factors, and comorbidities.

Mendelian randomization (MR) provides an effective and inexpensive alternative for estimating the potential effects of vitamin B12 levels in the general population. MR employs genetic variants as instruments for an exposure (i.e., vitamin B12 levels) and assesses the associations between the genetically predicted exposure and the outcomes under investigation[16,17]. MR relies on three instrumental variable assumptions: (1) the genetic instruments should be associated with the exposure; (2) the genetic instrument-outcome association should not be confounded; and (3) the genetic instruments should affect the outcome through the exposure and not through other mechanisms, which is known as the assumption of no horizontal pleiotropy. The first assumption can be fulfilled by selecting variants significantly associated with serum B12 levels as genetic instruments from recent large-scale genome-wide association studies (GWASs)[18]. The second and third assumptions require careful examination through sensitivity analyses. In particular, although the randomization of genetic variants at conception breaks associations with many conventional confounders, MR analyses, particularly those involving psychosocial traits, may still be biased by confounding due to uncontrolled demographic effects and indirect genetic effects[19]. These biases can be mitigated by using data from family-based studies[19–21].

MR has been adopted to evaluate the relationships between serum vitamin B12 levels and several diseases and biomarkers. Suggestive evidence has indicated that increased vitamin B12 levels may marginally increase fasting glucose while reducing beta-cell function in the pancreas[22]. However, no explicit associations have been identified between vitamin B12 levels and coronary artery disease, type 2 diabetes, measures of obesity, or Alzheimer's disease[22,23]. Recently, a phenome-wide study based on the UK Biobank confirmed the association between increased vitamin B12 levels and reduced odds of pernicious anemia[24]. However, it remains unclear whether the genetically predicted vitamin B12 levels are associated with psychiatric disorders and cognitive impairment, due to limited statistical power in previous studies.

Therefore, in this study, we performed MR analyses to investigate the potential effects of serum vitamin B12 levels on eight common psychiatric disorders and cognitive impairment in the general population. To maximize statistical power, we utilized the GWASs with the largest sample sizes to date, encompassing attention-deficit/hyperactivity disorder (ADHD), anorexia nervosa, anxiety disorders, autism spectrum disorder, bipolar disorder, depression, obsessive-compulsive symptoms (OCS), schizophrenia, as well as educational attainment and cognitive performance. Multiple sensitivity analyses were performed to address potential bias arising from violations of instrumental variable assumptions. This study aims to contribute to the understanding of whether vitamin B12 supplementation could serve as a preventive measure against psychiatric disorders and cognitive impairment in the general population.

## Methods
### Genome-wide association studies of serum vitamin B12 and folate levels
Total serum vitamin B12 levels were measured in three cohort studies[18]. GWAS meta-analyses were performed based on a total of 45,576 participants of European ancestry, including 37,283 individuals from an Icelandic cohort and 8293 individuals from two Danish cohorts. Genotyping in the

Danish cohorts focused on variants in protein-coding regions[18]. Furthermore, GWAS meta-analyses for serum folate levels were performed in the same cohort studies, based on a total of 37,341 individuals. Vitamin B12 and folate levels were quantile-normalized prior to analysis. Details of these GWAS meta-analyses have been described previously[18]. In total, 11 loci were significantly associated with serum vitamin B12 levels and two loci with folate levels ($p$-value $< 2.2 \times 10^{-9}$; Bonferroni-corrected genome-wide significance threshold accounting for 22.9 million tested variants)[18]. Lead variants at these loci with publicly available test statistics were used as genetic instruments. We calculated the $F$-statistic of each genetic instrument, where an $F$-statistic $>10$ was considered to indicate a low risk of weak instrument bias[25]. The genetic instruments and the mapped protein-coding genes are summarized in Supplementary Data 1.

### Known genotype-phenotype associations involving genetic instruments
To assess the potential risk of horizontal pleiotropy, we evaluated whether the selected genetic instruments have been associated with other phenotypes. We queried each genetic instrument in the Open Targets Genetics database[26,27] (https://genetics.opentargets.org/, accessed December 15, 2023), which includes existing GWASs in the GWAS Catalog as well as GWASs conducted using the UK Biobank or FinnGen resources[28–31]. After excluding associations with vitamin B12 levels, folate levels, and pernicious anemia, we defined genetic instruments that are associated with one to five phenotypes ($p$-value $< 5.0 \times 10^{-8}$) to be subject to a moderate risk of horizontal pleiotropy, and those associated with more than five phenotypes to be subject to a high risk of horizontal pleiotropy. This ad hoc classification was adopted to enable multi-layered sensitivity analyses, although it could not distinguish between horizontal and vertical pleiotropy. All genotype-phenotype associations involving the genetic instruments used in this study are summarized in Supplementary Data 2.

### Association between vitamin B12 levels and pernicious anemia
As a positive control, we assessed whether genetically predicted vitamin B12 and folate levels were associated with pernicious anemia. We obtained UK Biobank exome-based association statistics from the Genebass database[32] (https://app.genebass.org/, accessed October 19, 2023) for all genetic instruments available in the exome-sequencing data. Exome-sequencing data were prioritized over genotyping and imputation data because the instruments used for MR analyses are located in coding regions (Supplementary Data 1), where exome sequencing provides higher accuracy and direct detection of variants without relying on imputation[33]. Details regarding quality control, exome-side association studies, and data curation have been described previously[32]. We queried each genetic instrument for its association with pernicious anemia, based on the International Classification of Diseases version 10 (ICD-10) code of D51 or self-reported physician-made diagnoses. Additionally, we investigated associations with other types of anemia for comparison.

We performed MR to assess the association between the genetically predicted vitamin B12 levels and each type of anemia using the weighted median method due to the presence of pleiotropy in several genetic instruments (Supplementary Data 2). The weighted median method estimates the effect size of the exposure by calculating a weighted median of the Wald ratio estimates from individual genetic instruments, with weights corresponding to the inverse variance of each estimate. This method can provide robust estimates with up to 50% of the information coming from invalid genetic instruments[34]. We additionally performed MR using the inverse variance weighted regression, penalized weighted median, simple mode, weighted mode, and MR–Egger methods. Specifically, the inverse variance weighted regression method assumes that all instruments are valid[35]. The penalized weighted median method, similar to the weighted median method, assumes that the majority of the instruments are valid[34]. The simple mode and weighted mode methods assume that the plurality of the instruments are valid[36]. The MR-Egger method assumes that the strength of the instruments is independent of

their direct effects on the outcome[37]. A nominally significant MR-Egger intercept (p-value < 0.05) was considered evidence of directional pleiotropy[37]. We derived Wald ratio estimates to assess the association between the genetically predicted folate levels and each type of anemia, since only one genetic instrument for folate levels was available in the UK Biobank exome-sequencing data. MR analyses were conducted using the TwoSampleMR R package version 0.5.6[38].

## Genome-wide association studies of psychiatric disorders, educational attainment, and cognitive performance

The summary statistics of the most recent large-scale GWAS meta-analyses for eight psychiatric disorders were obtained from the Psychiatric Genomics Consortium, including ADHD[39] (N cases = 20,183, N controls = 35,191), anorexia nervosa[40] (N cases = 16,992, N controls = 55,525), anxiety disorders[41] (N = 1,096,458), autism spectrum disorder[42] (N cases = 18,381, N controls = 27,969), bipolar disorder[43] (N cases = 41,917, N controls = 371,549), depression[44] (N cases = 550,355, N controls = 2,071,918), OCS[45] (N = 33,943), and schizophrenia[46] (N cases = 69,369, N controls = 236,642). The GWAS summary statistics for educational attainment[47] (N = 765,283), measured as the number of years of schooling completed, were obtained from the Social Science Genetic Association Consortium. The GWAS summary statistics for cognitive performance[48] (N = 257,828), quantified based on various cognitive tests, were obtained from the Cognitive Genomics Consortium. Participants of these study cohorts were predominantly of European ancestry. These GWASs are summarized in Table 1. The samples included in the exposure GWASs partially overlapped with those in the outcome GWASs for bipolar disorder, depression, schizophrenia, and educational attainment. However, the expected bias toward observational estimates was minimal given the large F-statistics of the instruments[49] (Supplementary Data 1). For the other outcome GWASs, there was no known sample overlap.

## Power calculation for Mendelian randomization

To assess the likelihood of type II errors, we conducted power calculations for MR analyses using the mRnd R package[50] (https://shiny.cnsgenomics.com/mRnd/, accessed December 15, 2023). For each outcome, we calculated the required true effect size associated with a one standard deviation change in vitamin B12 levels to achieve 80% power, with a type I error rate (i.e., significance level) of 5% and 0.25% (Bonferroni-corrected significance threshold accounting for two exposures and ten outcomes), respectively. The effect size was expressed as an odds ratio (OR) for binary outcomes and as a standard deviation change for quantitative outcomes.

## Associations between vitamin B12 levels and psychiatric disorders, educational attainment, and cognitive performance

Similar to positive control analyses, we performed MR to assess the association between the genetically predicted vitamin B12 levels and each of the outcomes. MR estimates derived using the weighted median method were reported as primary results. Associations with a p-value < $2.5 \times 10^{-3}$ (Bonferroni-corrected significance threshold accounting for two exposures and ten outcomes) were considered significant. Secondary analyses were performed using the inverse variance weighted regression, penalized weighted median, simple mode, weighted mode, and MR–Egger method, respectively. For folate levels, since two genetic instruments were used, the inverse variance weighted regression estimates were derived to assess the association with each outcome.

## Sensitivity analyses excluding pleiotropic genetic instruments

To mitigate potential bias from horizontal pleiotropy, we excluded genetic instruments that were subject to a high risk of horizontal pleiotropy and repeated the MR analyses for vitamin B12 levels. Subsequently, we further excluded genetic instruments that were subject to a moderate risk of horizontal pleiotropy and again repeated the MR analyses for vitamin B12 levels.

**Table 1 | Genome-wide association studies of psychiatric disorders, educational attainment, and cognitive performance**

| Outcome | Participating cohorts | Cases | Controls | Total |
|---|---|---|---|---|
| Attention-deficit/hyperactivity disorder | A Danish cohort collected by the Lundbeck Foundation Initiative for Integrative Psychiatric Research, and 11 European, North American and Chinese cohorts aggregated by the Psychiatric Genomics Consortium | 20,183 | 35,191 | 55,374 |
| Anorexia nervosa | 13 European ancestry cohorts from 17 countries, including the Anorexia Nervosa Genetics Initiative and the Eating Disorders Working Group of the Psychiatric Genomics Consortium | 16,992 | 55,525 | 72,517 |
| Anxiety disorders[a] | Five European ancestry cohorts from Europe and North America, and the Psychiatric Genomics Consortium | | | 1,096,458 (Effective sample size = 529,764) |
| Autism spectrum disorder | A Danish cohort collected by the Lundbeck Foundation Initiative for Integrative Psychiatric Research | 18,381 | 27,969 | 46,350 |
| Bipolar disorder | 57 predominantly European ancestry cohorts collected in Europe, North America and Australia | 41,917 | 371,549 | 413,466 |
| Depression | 105 diverse cohorts, including ~75% European ancestry participants | 550,355 | 2,071,918 | 2,622,273 |
| Obsessive-compulsive symptoms[b] | Seven European ancestry cohorts collected in Europe and North America | | | 33,943 |
| Schizophrenia | 91 cohorts, including ~80% European ancestry participants and ~20% East Asian ancestry participants | 69,369 | 236,642 | 306,011 |
| Educational attainment[c] | 69 European ancestry cohorts in the Social Science Genetic Association Consortium and the UK Biobank European ancestry individuals | | | 765,283 |
| Cognitive performance[d] | 24 European ancestry cohorts in the Cognitive Genomics Consortium and the UK Biobank European ancestry individuals | | | 257,828 |

[a]Meta-analysis of five case-control studies and the Million Veteran Program, which measured anxiety using the generalized anxiety disorder two-item scale score; effective sample sizes for case-control studies were calculated as $\frac{4 \times N_{Case} \times N_{Control}}{N_{Case} + N_{Control}}$.
[b]Obsessive-compulsive symptoms were measured by standardized obsessive-compulsive item scores.
[c]Educational attainment was measured as the number of years of schooling completed.
[d]Cognitive performance was quantified based on various tests conducted in each study cohort, including multiple neuropsychological or intelligence quotient tests in the Cognitive Genomics Consortium and a test of verbal-numerical reasoning in the UK Biobank.

## Within-sibship genome-wide association studies

Since population GWAS estimates for psychiatric disorders, educational attainment, and cognitive performance may be confounded by uncontrolled demographic effects and indirect genetic effects, we repeated the MR analyses leveraging within-sibship GWAS estimates that were available for depressive symptoms, educational attainment, and cognitive performance[19]. Notably, unlike in the GWAS for depression, depressive symptoms were measured using various rating scales and standardized to be a continuous outcome[19]. European ancestry-specific within-sibship GWAS summary statistics were obtained from the Within Family Consortium[19]. Details of these GWASs have been described previously and are summarized in Supplementary Data 3.

## Reporting summary

Further information on research design is available in the Nature Portfolio Reporting Summary linked to this article.

## Results

### Mendelian randomization confirmed the association between vitamin B12 levels and pernicious anemia

An overview of this study is illustrated in Supplementary Fig. 1. Meta-analyses of GWASs for serum vitamin B12 levels identified 11 genetic instruments ($p$-value $< 2.2 \times 10^{-9}$; Supplementary Data 1). All 11 instruments are either missense variants, which result in a different amino acid in the protein, or stop-gain variants, which lead to truncated proteins, affecting protein-coding genes (Supplementary Data 1). Of these genetic instruments, the minimal F-statistic was 43.14, suggesting a low risk of weak instrument bias (Supplementary Data 1). Notably, some of these genetic instruments may pose a risk of horizontal pleiotropy. For instance, rs602662, a missense variant of *FUT2*, was subject to a high risk of horizontal pleiotropy (Supplementary Data 2). Missense variants in *MUT*, *FUT6*, *CD320*, and *CUBN* were subject to a moderate risk of horizontal pleiotropy due to their associations with other phenotypes, such as liver function biomarkers, lipid levels, and height (Supplementary Data 2).

Based on the UK Biobank, we confirmed that a one standard deviation increase in genetically predicted vitamin B12 levels was strongly associated with an OR of 0.27 for vitamin B12 deficiency anemia based on ICD-10 codes (95% CI: 0.19–0.40; $p$-value $= 1.0 \times 10^{-11}$), as well as an OR of 0.24 for self-reported pernicious anemia (95% CI: 0.15–0.40; $p$-value $= 2.1 \times 10^{-8}$; Fig. 1 and Supplementary Data 4). Meanwhile, the genetically predicted vitamin B12 levels were not associated with other types of anemia (Fig. 1 and Supplementary Data 4). MR estimates derived using different methods were highly consistent (Supplementary Data 4).

In addition, two independent genetic variants, rs1801133 and rs652197, were significantly associated with serum folate levels ($p$-value $< 2.2 \times 10^{-9}$; Supplementary Data 1). However, rs1801133, a missense variant of *MTHFR*, demonstrated a high risk of horizontal pleiotropy, with known associations with N-terminal prohormone brain natriuretic peptide levels, blood pressure, and multiple blood cell characteristics (Supplementary Data 2). Meanwhile, rs652197, an intronic variant of *FOLR3*, has been associated with serum 25-hydroxyvitamin D levels in previous studies (Supplementary Data 2). As expected, the genetically predicted folate levels were also associated with a reduced odds of vitamin B12 deficiency anemia based on ICD-10 codes (OR = 0.30; 95% CI: 0.18–0.49; $p$-value $= 2.3 \times 10^{-6}$) and self-reported pernicious anemia (OR = 0.29; 95% CI: 0.14–0.60; $p$-value $= 9.3 \times 10^{-4}$; Fig. 1 and Supplementary Data 4). Although folate levels were predicted to reduce the odds of folate deficiency anemia, this association was not significant, likely due to the small number of folate deficiency anemia cases (Fig. 1 and Supplementary Data 4). Given the risk of horizontal pleiotropy in the instruments for folate levels and the inapplicability of MR sensitivity analysis methods, subsequent analyses and interpretation of findings did not focus on folate levels.

### Genetically predicted vitamin B12 levels were not associated with psychiatric disorders, educational attainment, or cognitive performance

The large-scale GWASs for psychiatric disorders, educational attainment, and cognitive performance ensured sufficient statistical power in MR analyses (Table 1 and Supplementary Data 5). Of the eight psychiatric disorders under investigation, the GWAS meta-analyses for depression had the largest sample size. As a result, with a type I error rate of 0.25% (Bonferroni-corrected significance threshold), MR analyses with depression as the outcome could achieve 80% power when the true OR per one standard deviation change in vitamin B12 levels was ≥1.03 (Supplementary Data 5). For educational attainment and cognitive performance, MR analyses could achieve 80% power when the true effect sizes were ≥0.020 and ≥0.034 per one standard deviation change in vitamin B12 levels, respectively (Supplementary Data 5).

Despite sufficient statistical power, MR analyses did not detect any significant associations between genetically predicted vitamin B12 levels and any outcomes (Fig. 2 and Supplementary Data 5). For example, a one standard deviation increase in genetically predicted vitamin B12 levels was associated with an OR of 1.02 for depression (95% CI: 1.00–1.04; $p$-value = 0.11), as well as a 0.0077 and 0.013 standard deviation increase in educational attainment (95% CI: −0.010 to 0.025; $p$-value = 0.39) and cognitive performance (95% CI: −0.0088 to 0.035; $p$-value = 0.24),

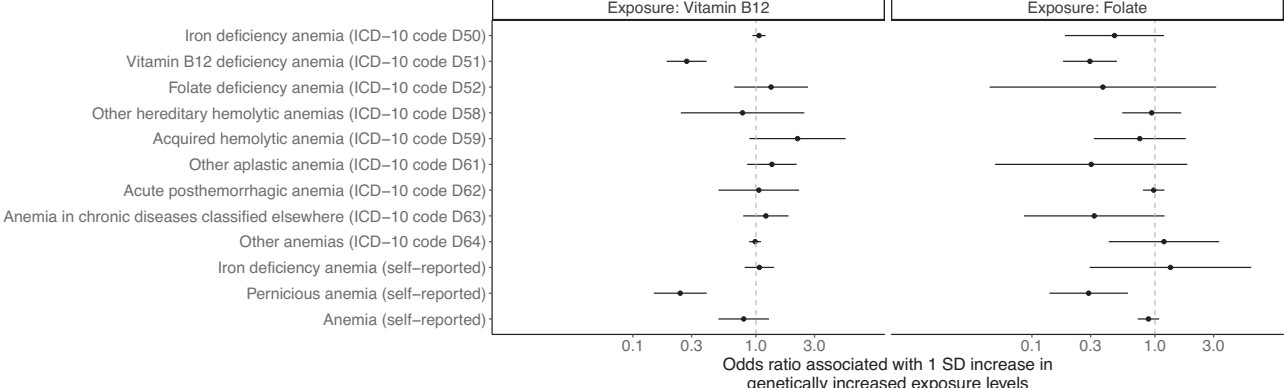

**Fig. 1 | Associations between genetically predicted vitamin B12 and folate levels and various types of anemia.** Mendelian randomization estimates obtained using the weighted median method and the Wald ratio method are illustrated for vitamin B12 levels and folate levels, respectively. Error bars represent 95% confidence intervals. Sample sizes are available in Supplementary Data 4.

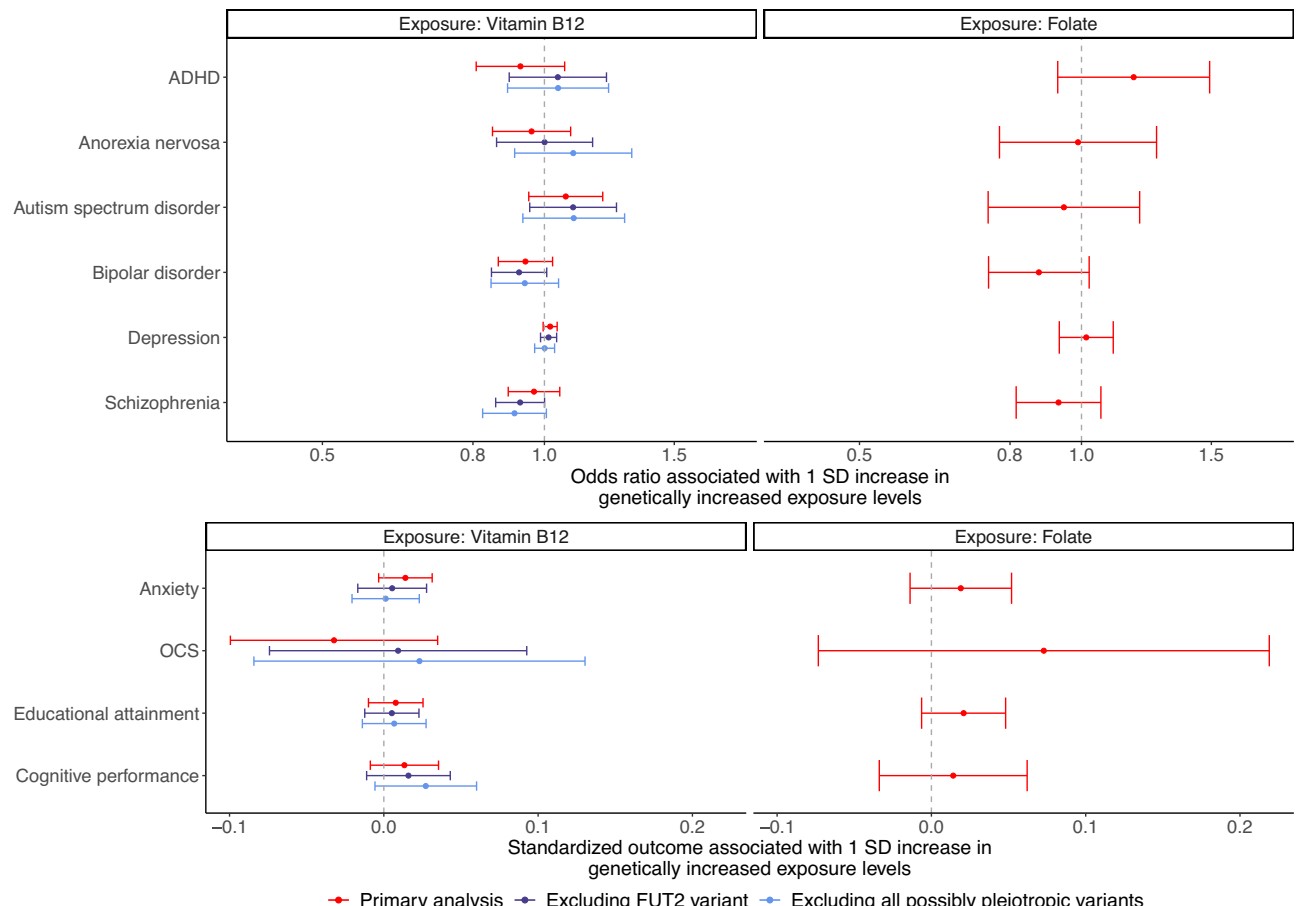

**Fig. 2 | Associations between genetically predicted vitamin B12 and folate levels and eight psychiatric disorders, educational attainment, and cognitive performance.** Sensitivity analyses for vitamin B12 levels were performed after excluding genetic instruments that were potentially subject to horizontal pleiotropy.

Mendelian randomization estimates obtained using the weighted median method and the inverse variance weighted regression method are illustrated for vitamin B12 levels and folate levels, respectively. Error bars represent 95% confidence intervals. Sample sizes are available in Table 1.

respectively (Fig. 2 and Supplementary Data 5). The estimated associations between folate levels and the outcomes also largely overlapped with the null (Fig. 2 and Supplementary Data 6).

Furthermore, no associations were detected between genetically predicted vitamin B12 levels and any outcomes in sensitivity analyses where the genetic instruments demonstrating potential horizontal pleiotropy were excluded (Fig. 2 and Supplementary Data 7 and 8). In fact, none of the tested associations had a nominal $p$-value < 0.05 in the primary analyses using the weighted median method (Fig. 2). The estimates derived using alternative methods were highly consistent with the primary results, with the only exception that with the penalized weighted median method, a one standard deviation increase in genetically predicted vitamin B12 levels was associated with an OR of 0.82 for ADHD (95% CI: 0.72–0.93; $p$-value = $2.1 \times 10^{-3}$; Supplementary Fig. 2 and Supplementary Data 5). Nevertheless, this association was not detected after excluding the highly pleiotropic genetic instrument in *FUT2* (Supplementary Data 7) as well as after further excluding the moderately pleiotropic genetic instruments (Supplementary Data 8).

### Sensitivity analyses based on within-sibship genome-wide association studies

MR analyses leveraging within-sibship GWASs for depressive symptoms, educational attainment, and cognitive performance did not identify any significant associations with genetically predicted vitamin B12 levels (Fig. 3A). Specifically, a one standard deviation increase in genetically predicted vitamin B12 levels was associated with a 0.054 standard deviation

decrease in depressive symptoms (95% CI: −0.14 to 0.036; $p$-value = 0.24), as well as a 0.0056 and 0.039 standard deviation increase in educational attainment (95% CI: −0.043 to 0.055; $p$-value = 0.82) and cognitive performance (95% CI: −0.066 to 0.14; $p$-value = 0.47), respectively (Fig. 3A and Supplementary Data 9). These estimates were consistent with those obtained using alternative MR methods (Fig. 3B–D and Supplementary Data 9).

### Discussion

Vitamin B12 has important roles in one-carbon metabolism as both an enzyme cofactor or substrate[1]. Individuals with vitamin B12 deficiency can develop pernicious anemia as well as neuropsychiatric diseases[7,8]. However, it remains unclear whether the general population may benefit from vitamin B12 supplementation for protection against psychiatric disorders and cognitive impairment. In this study, we performed MR analyses to estimate the potential effects of total serum vitamin B12 levels on eight psychiatric disorders, educational attainment, and cognitive performance. We did not detect any significant associations between genetically predicted serum vitamin B12 levels and any of these outcomes.

Our study has several strengths. First, we employed genetic instruments that were protein-altering variants affecting crucial genes in the absorption, transport, or enzymatic reactions of vitamin B12 for MR analyses, which effectively guarded against weak instrument bias (Supplementary Data 1). Second, the use of large-scale GWASs for the outcomes, particularly depression, educational attainment, and cognitive performance, minimized the likelihood of type II errors, given the reporting of null

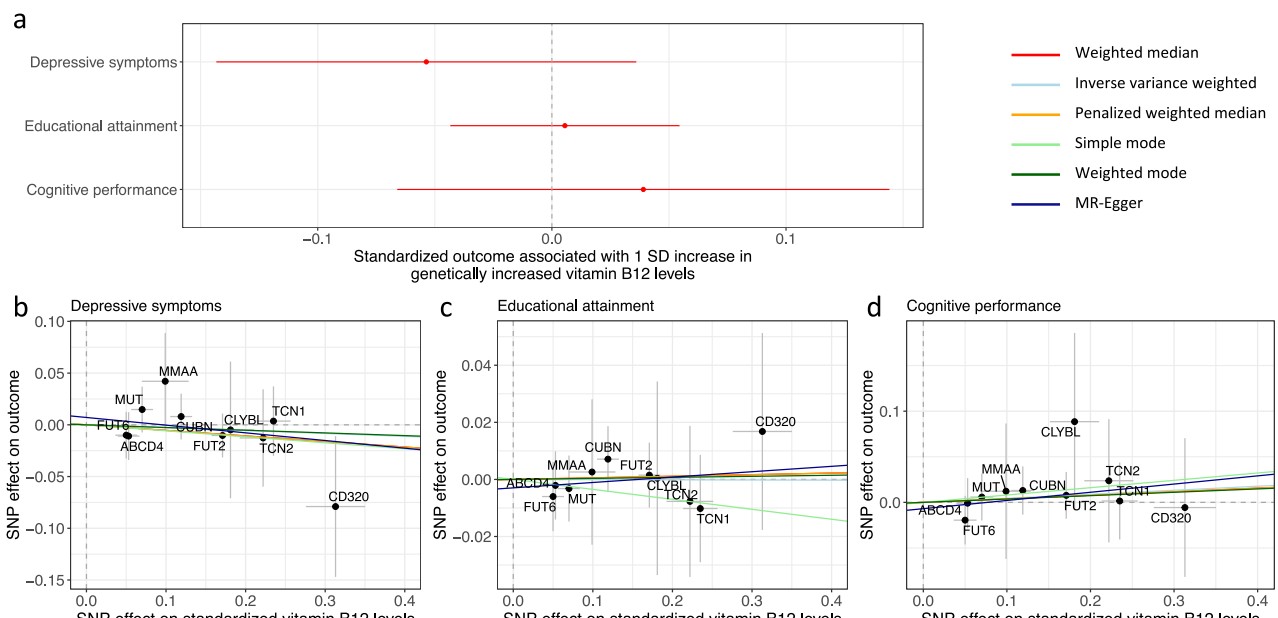

**Fig. 3 | Estimated effects of vitamin B12 levels based on within-sibship studies. a** Associations between genetically predicted vitamin B12 and depressive symptoms, educational attainment, and cognitive performance. Mendelian randomization estimates obtained using the weighted median method are illustrated. Scatter plots comparing the genetic instrument-outcome and the genetic instrument-vitamin B12 levels associations are illustrated for **b** depressive symptoms, **c** educational attainment, and **d** cognitive performance. The mapped genes of the respective genetic instruments are indicated. The slopes of the colored lines represent estimates obtained through different Mendelian randomization methods. Error bars represent 95% confidence intervals. Sample sizes are available in Supplementary Data 3.

associations. These results were compared to the positive control, where genetically increased vitamin B12 levels exhibited strong protective effects against pernicious anemia, thereby supporting the validity of our analyses. Third, consistent MR estimates were obtained in sensitivity analyses where pleiotropic variants were removed, safeguarding an indispensable instrumental variable assumption of MR. Nevertheless, MR estimates for folate levels were more prone to bias due to horizontal pleiotropy; thus, interpreting the results involving folate levels requires extra caution. Fourth, MR analyses based on within-sibship GWASs for depressive symptoms, educational attainment, and cognitive performance further reduced the risk of confounding due to uncontrolled population stratification, assortative mating, or indirect genetic effects[19,51].

Our study has a clear and important implication that, in the general population, vitamin B12 supplementation is unlikely to meaningfully reduce the risks of the investigated psychiatric disorders or significantly improve educational attainment or cognitive performance. These results may discourage randomized controlled trials amongst individuals without clinical vitamin B12 deficiency, while encouraging the identification of other risk factors and preventive measures for psychiatric disorders and cognitive impairment. However, it is noteworthy that none of the GWASs utilized in this study were based on populations selected for vitamin B12 deficiency. Importantly, our findings should not be interpreted as refuting any known biological functions of vitamin B12, particularly in individuals with deficiency or specific medical conditions where supplementation is clinically indicated. Multiple lines of evidence are still needed to ascertain the potential impact of vitamin B12 or folate plus vitamin B12 supplementation on various outcomes in individuals with vitamin B12 deficiency.

Some limitations of our study should be noted. First, our MR analyses were based on total serum vitamin B12 levels rather than the bioactive form, which accounts for approximately 20% of circulating vitamin B12[3,4]. Further research could explore associations using bioactive vitamin B12 and incorporate triangulation with observational evidence as more data become available. Second, we relied on sex-combined GWASs and could not identify potential sex-specific effects of vitamin B12 levels, since large-scale sex-stratified GWASs for both vitamin B12 levels and the outcomes are yet to emerge. Third, our analyses were restricted to populations of European

ancestry. It remains unclear whether our findings could be generalized to other populations of non-European ancestries. Fourth, our MR analyses could estimate population-averaged associations between vitamin B12 levels and the outcomes, but not the potential dose-dependent effects of vitamin B12 levels. Furthermore, the distribution of raw vitamin B12 levels may be skewed, which may result in inconsistent interpretation of a one-standard-deviation change across the population. Investigating these aspects would require availability of both vitamin B12 measurements and the outcomes in the same study population[52,53]. Fifth, one of the genetic instruments, rs371753672, a missense variant of *MMACHC*, was not available in any GWASs utilized in this study due to its low minor allele frequency. However, this variant only captures a small proportion of the variance in vitamin B12 levels. Further research, such as larger GWAS for vitamin B12 levels with publicly available summary statistics, may help identify additional instruments for MR analyses, potentially allowing for a more lenient instrument selection approach and increasing the statistical power to detect potential associations. Last, our results do not provide information on whether serum vitamin B12 levels may influence the progression of psychiatric disorders or cognitive impairment, as the genetic architecture underlying disease progression may differ from that underlying disease pathogenesis. We anticipate that future investigations will more comprehensively illuminate the role of vitamin B12 in various psychiatric disorders and cognitive impairment in diverse populations.

In conclusion, through MR analyses, we demonstrated that genetically predicted total serum vitamin B12 levels were not associated with eight psychiatric disorders, educational attainment, or cognitive performance. Our findings indicate that vitamin B12 supplementation is unlikely to offer clinically meaningful protection against these psychiatric disorders or cognitive impairment in the general population.

## Data availability

GWAS summary statistics are available from the Psychiatric Genomics Consortium (https://pgc.unc.edu/). All data generated or analyzed during this study are included in this manuscript and its supplementary information files. The numerical results underlying the graphs presented in the main figures are available in Supplementary Data 4–9.

## Code availability

Code for all analyses is available at the Figshare repository: https://doi.org/10.6084/m9.figshare.25852813.v1

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

## Acknowledgements

We thank David S. Rosenblatt and David Watkins for helpful discussions. T.L. has been supported by a Schmidt AI in Science Postdoctoral Fellowship and start-up funding from the Office of the Vice Chancellor for Research and Graduate Education, School of Medicine and Public Health, and Department of Population Health Sciences at the University of Wisconsin-Madison. The funders have no role in study design; collection, management, analysis and interpretation of data; or the decision to submit for publication.

## Author contributions

T.L. and A.D.P. conceptualized the study. T.L. curated the data, performed the analyses, and wrote the original draft. T.L. and A.D.P. interpreted the results and reviewed and edited the paper critically.

## Competing interests

T.L. has been providing consulting services to 5 Prime Sciences Inc., which was not involved in the design, execution, analysis, or interpretation of the study. The remaining authors declare no competing interests.

## Inclusion & Ethics

This study used summary statistics from previously published genome-wide association studies, which individually obtained ethics approval and consent to participate.

## Additional information

[1]Department of Population Health Sciences, School of Medicine and Public Health, University of Wisconsin-Madison, Madison, WI, USA. [2]Department of Biostatistics and Medical Informatics, School of Medicine and Public Health, University of Wisconsin-Madison, Madison, WI, USA. [3]Center for Demography of Health and Aging, University of Wisconsin-Madison, Madison, WI, USA. [4]Center for Genomic Science Innovation, University of Wisconsin-Madison, Madison, WI, USA. [5]Center for Human Genomics and Precision Medicine, University of Wisconsin-Madison, Madison, WI, USA. [6]Department of Statistical Sciences, Faculty of Arts and Science, University of Toronto, Toronto, ON, Canada. [7]Division of Biostatistics, Dalla Lana School of Public Health, University of Toronto, Toronto, ON, Canada. [8]Division of Epidemiology, Dalla Lana School of Public Health, University of Toronto, Toronto, ON, Canada. [9]Genetics and Genome Biology, The Hospital for Sick Children, Toronto, ON, Canada. ✉e-mail: tianyuan.lu@wisc.edu; andrew.paterson@sickkids.ca

