## [Transparent Peer Review file · Communications Medicine]

Estimating effects of serum vitamin B12 levels on psychiatric disorders and cognitive impairment: a Mendelian randomization study

Corresponding Author: Dr Tianyuan Lu

Version 0:

Reviewer comments:

Reviewer #1

(Remarks to the Author)

The authors investigated the associations of genetically predicted serum vitamin B12 with psychiatric disorders and cognitive functions. I have the following suggestions for consideration.

1. [Line 76-80] I am skeptical about the fulfilment of the independence assumption. For example, population stratification may confound the associations of genetic variants with the outcome and violate this assumption.
2. [Line 105] The authors investigated both serum vitamin B12 and folate. I suggest the authors clarify the two exposures throughout the manuscript.
3. [Line 143-144] Could the authors explain why the weighted median rather than IVW was used as the main analysis?
4. [Line 151-152] Could the authors explain why only instruments available in the UK Biobank exome-sequencing data were used?
5. [Line 166] Could the authors explain more about the measures of cognitive performance?
6. [Line 185] Given that the authors included two exposures, the Bonferroni-corrected significance threshold should be $0.05/(2*10)$.
7. [Line 192-195] Could the authors clarify their definition of high and moderate risk of horizontal pleiotropy?
8. [Supplementary Table S4] Did the authors use only pernicious anemia or all types of anemia as positive control outcomes? Please explain the selection of positive control outcomes in the method section.
9. [Supplementary Tables S5-6] Could the authors report the true effect size for 80% power rather than several columns of power with differing true effects? Please also make relevant changes in the Methods and Results sections.
10. [Discussion] Could the authors compare their findings with previous evidence from trials and observational studies? Could the authors provide more biological explanations of their findings?
11. [Line 297-299] The authors selected genetic variants significantly associated with serum vitamin B12 and folate levels with a p-value $< 2.2 \times 10^{-9}$, which is a statistical approach. Could the authors explain more about their crucial roles in the absorption, transport, or enzymatic reactions of vitamin B12 and why it is relevant to unmeasured confounding and reverse causation?
12. [Line 307] Could the authors explain why the MR estimates for folate level are more prone to horizontal pleiotropy?

Reviewer #2

(Remarks to the Author)

Lu and Paterson conducted an MR study to explore the association between serum vitamin B12 levels and psychiatric disorders and cognitive impairment. They found no association between genetically predicted serum vitamin B12 levels and either of the eight psychiatric disorders, educational attainment or cognitive performance. Even though the analysis was well performed, these null associations were reported in previous studies, like some studies cited by the authors, a recent PheWAS on B vitamins (pmid: 36811473), and this MR study (pmid: 38999734). These findings provide limited novelty.

Reviewer #3

(Remarks to the Author)

Lu et al. explored associations of serum vitamin B12 with psychiatric disorders and cognitive impairment using Mendelian randomization (MR). My specific comments are below:

1. Line 74, in terms of the 2nd MR assumption, please specify the confounders.
2. Please add suitable sensitivity analysis for folate, as it only has two genetic instruments.
3. In line 180 – 189, please briefly summarize how each MR sensitivity analysis deals with potential violations of MR assumption(s).
4. Please also explain why weighted median approach was used as the main results.
5. GWAS of depression (i.e. reference 38) is out-of-date. Please consider the most recent one.
6. Lines 192 – 193, please explain how to discriminate horizontal pleiotropy from vertical pleiotropy.
7. This study is a two-sample MR. Please comment on whether there is a sample overlap. If there is no sample overlap, then it would be a strength.
8. Please add a DAG to explain MR based on within-sibship GWAS in Figure 1.
9. The numbers of cases for anxiety disorders and obsessive-compulsive disorder are insufficient. Please consider meta-analyzing the GWAS currently used with other publicly available GWAS (e.g. from FinnGen) of the same outcomes.

Reviewer #4

(Remarks to the Author)

It is my pleasure to review this manuscript. The authors applied comprehensive Mendelian randomization methods to explore the effect of vitamin B12 on eight psychiatric disorders and found no effect of vitamin B12 on any of these disorders. It is a well conducted study. Although it suggests null results, it is still worthy to be addressed in the nutritional epidemiology and has an insight in clinical practice. On top of these, there are some concerns from my end, which might be additional before going to an official publication:

Major:

- The exome-sequencing data focus on protein-coding regions within the genome. It represents only about 1-2% of the entire genome but contain the majority of disease-causing variants. For the positive control of pernicious anemia, can the author clarify why only the exome-sequencing data is applied? Are there any general GWAS data available? If so, why please clarify why the general GWAS data is not applied. I imagine the exome-sequencing data for the other 8 outcomes are not available?
- Please clarify what are missense or stop-gain variants. It is not friendly to the audience who are not familiar with this advanced genetic terminology. I believe this is one of the reasons why you apply exome-sequencing GWAS. Besides, any general GWAS for vitamin B12 is available? Though the genetic variants obtained from general GWAS are likely subject to the violations of the IV assumptions. But I think this can be briefly discussed.
- Although the observational design is open to confounding, is it available to conduct an observational design in this study? Triangulation of the estimates would be appreciated.
- I agree that it is important to account for the potential pleiotropy in an MR design. But the standard of how to define a “moderate” risk of pleiotropy is not clear. For example, “Missense variants in MUT, FUT6, CD320, and CUBN were subject to a moderate risk of horizontal pleiotropy due to their associations with other phenotypes, such as liver function biomarkers, lipid levels, and height.” It is not convincing to me that these phenotypes are causally associated with psychiatric disorders. I appreciate the way of excluding the potential pleiotropy SNPs to account for the violation of the exclusion restriction assumption. But the exclusion criteria should be clear and with scientific evidence (e.g., relatively clear evidence of being a causal phenotype of the outcome of interest).
- If I understand correctly, Figure 2 is missing. Figure 3 in the manuscript is Figure 2 and Figure 4 in the manuscript is Figure 3.

Minor:

- It would be more straightforward that unit of presenting the point estimate in regular format (e.g., a 0.008 in SD unit, 95% CI: -0.010 to 0.025, p-value = 0.39) and make it consistent with the number of decimal place.
- Line 126, it is not clear to me what are these five other phenotypes.
- Line 173, I imagine the SD unit in the power calculation is based on 5% type I error and 80% power.
- Please forgive this is a subjective comment, I don't think Figure 1 is not very helpful (e.g., not clear in terms of study design and/or flow) for the audiences to have a quick understanding of what this study is about and how it is conducted.
- Line 231, “rs1801133, a missense variant of MTHFR, demonstrated a high risk of horizontal pleiotropy”, what are the phenotypes of the high risk of horizontal pleiotropy?
- I appreciate that the author has done lots of work on the power calculation. For clinical / public health perspective, I would suggest the author assign/set up a “clinical effect size” (like a benchmark), which can be based on the clinical and public health background/information (e.g., combining the prevalence of the disease outcome and if an effective medical treatment is available, an ideal effect size e.g., OR = 1.01 would be, for example, 10% cases of the disease can be prevented in the population. Therefore, we consider this a reference to discuss whether the MR estimate we obtained has enough power). In this case, it provides an intuitive understanding of where a practical effect size would be.
- Is it possible to access how many pmol per litre as indicated by 1 SD unit? Also, what is the distribution of Vitamin B12 level in the population? If the Vitamin B12 is skewed distributed, a 1SD change is not ideal to interpret the results, which can be acknowledged in the discussion.

Version 1:

Reviewer comments:

Reviewer #1

(Remarks to the Author)

I commend the authors for their thorough clarification of this study. I have follow-up questions about instrument selection.

[Line 120] How many independent genetic variants had a p-value $< 2.2 \times 10^{-9}$ before restricting to exome-sequencing data? What is the r^2 for excluding correlated genetic variants? The analyses could have limited power because many non-coding variants were excluded. Also, Bonferroni-corrected threshold accounting for 22.9 million tested variants is not appropriate because not all the tested variants are independent.

Overall, I think the authors applied a very stringent approach for instrument selection. Given this study reported null findings, I would recommend sensitivity analyses using a more lenient and common approach for instrument selection, i.e., independent ($r^2 < 0.001$) genetic variants with p value $< 5 \times 10^{-8}$ without restricting to coding variants.

Reviewer #3

(Remarks to the Author)

I appreciate the authors' efforts to amend the manuscript, and do not have any further comments.

Reviewer #4

(Remarks to the Author)

I am happy with the author's responses and the corresponding revision which address most of my concerns.

Version 2:

Reviewer comments:

Reviewer #1

(Remarks to the Author)

The authors have addressed my concerns.

Reviewer #1 (Remarks to the Author):

The authors investigated the associations of genetically predicted serum vitamin B12 with psychiatric disorders and cognitive functions. I have the following suggestions for consideration.

-We appreciate comments and suggestions from Reviewer 1.

1. [Line 76-80] I am skeptical about the fulfilment of the independence assumption. For example, population stratification may confound the associations of genetic variants with the outcome and violate this assumption.

-We thank Reviewer 1 for raising this important point and apologize for any confusion in our previous narrative. We fully agree that MR analyses may be subject to confounding due to uncontrolled demographic effects (e.g., population stratification and assortative mating) and indirect genetic effects, particularly for psychosocial traits. This is precisely why we conducted additional sensitivity analyses using within-sibship GWASs, which, despite their relatively small sample sizes, can effectively mitigate—though not entirely eliminate—potential bias from residual confounding. The consistency of our findings, with no significant associations observed in both population-based and within-sibship GWASs, further strengthens the credibility of our results. We have revised in Introduction:

“In particular, although the randomization of genetic variants at conception breaks associations with many conventional confounders, MR analyses, particularly those involving psychosocial traits, may still be biased by confounding due to uncontrolled demographic effects and indirect genetic effects¹. These biases can be mitigated by using data from family-based studies¹⁻³.”

2. [Line 105] The authors investigated both serum vitamin B12 and folate. I suggest the authors clarify the two exposures throughout the manuscript.

-We appreciate this comment. Indeed, our analyses involved both vitamin B12 and folate levels. The findings pertaining to serum folate levels are reported in Results:

“As expected, the genetically predicted folate levels were also associated with a reduced odds of vitamin B12 deficiency anemia based on ICD-10 codes (OR = 0.30; 95% CI: 0.18 – 0.49; p-value = 2.3×10^{-6}) and self-reported pernicious anemia (OR = 0.29; 95% CI: 0.14 – 0.60; p-value = 9.3×10^{-4} ; **Figure 1 and Supplementary Table S4**).”

“The estimated associations between folate levels and the outcomes also largely overlapped with the null (**Figure 2 and Supplementary Table S6**).”

However, we recognize important limitations of the analyses involving serum folate levels. First, the instruments for serum folate levels demonstrated pleiotropy (please also refer to our response to point #12). Second, since there were only two instruments for serum folate levels, existing horizontal pleiotropy-robust MR methods could not be applied, which all require at least three instruments. Therefore, our manuscript focused on serum vitamin B12 levels rather than folate levels, as explained in Results and Discussion:

“Given the risk of horizontal pleiotropy in the instruments for folate levels and the inapplicability of MR sensitivity analysis methods, subsequent analyses and interpretation of findings did not focus on folate levels.”

“Nevertheless, MR estimates for folate levels were more prone to bias due to horizontal pleiotropy, thus interpreting the results involving folate levels requires extra caution.”

3. [Line 143-144] Could the authors explain why the weighted median rather than IVW was used as the main analysis?

-This decision was made because several instruments demonstrated significant associations with multiple traits (**Supplementary Table S2**). While distinguishing between horizontal and vertical pleiotropy remains challenging, we chose the weighted median method for our primary analysis due to its greater robustness to invalid instruments compared to the IVW method, which assumes all instruments are valid. We also applied the IVW method and several alternative approaches (**Supplementary Table S5**), all of which yielded consistent results showing no significant associations. We have clarified in Methods:

“We performed MR to assess the association between the genetically predicted vitamin B12 levels and each type of anemia using the weighted median method due to the presence of pleiotropy in several genetic instruments (**Supplementary Table S2**). The weighted median method estimates the effect size of the exposure by calculating a weighted median of the Wald ratio estimates from individual genetic instruments, with weights corresponding to the inverse variance of each estimate. This method can provide robust estimates with up to 50% of the information coming from invalid genetic instruments⁴. We additionally performed MR using the inverse variance weighted regression, penalized weighted median, simple mode, weighted mode, and MR-Egger methods. Specifically, the inverse variance weighted regression method assumes that all instruments are valid⁵.”

4. [Line 151-152] Could the authors explain why only instruments available in the UK Biobank exome-sequencing data were used?

-We thank Reviewer 1 for this question. We used the exome-sequencing data because all of the instruments are missense or stop-gain variants in the coding regions, for which the exome-sequencing data provide higher accuracy and direct detection of variants without relying on imputation. We have clarified in Methods:

“Exome-sequencing data were prioritized over genotyping and imputation data because the instruments used for MR analyses are located in coding regions (**Supplementary Table S1**), where exome sequencing provides higher accuracy and direct detection of variants without relying on imputation⁶.”

To ensure the validity of our results, we repeated the MR analyses using GWAS summary statistics from the Pan-UK Biobank⁷, which were derived from genotyping and imputation data. The positive control associations with pernicious anemia remained significant and showed consistent effect magnitudes compared to those obtained using whole-exome sequencing data (**Additional Figures 1 and 2**). For conciseness, these additional analyses are not included in the main manuscript.

Additional Figure 1. Associations between genetically predicted vitamin B12 levels and various types of anemia based on Pan-UK Biobank GWASs.

Additional Figure 2. Associations between genetically predicted vitamin B12 levels and various types of anemia based on Pan-UK Biobank GWASs.

5. [Line 166] Could the authors explain more about the measures of cognitive performance?

-The measures of cognitive performance involved various study-specific tests that were detailed in the meta-analysis of GWASs⁸ and cohort studies. We have included in the footnote of Table 1:

“Cognitive performance was quantified based on various tests conducted in each study cohort, including multiple neuropsychological or intelligence quotient tests in the Cognitive Genomics Consortium and a test of verbal-numerical reasoning in the UK Biobank.”

Here, we provide the original description from the literature (Supplementary Table S40 from Lee et al.⁸):

(UK Biobank) “Standardized score on a test of verbal-numerical reasoning (data field 20016 for in-person assessments and data field 20191 for an online follow-up). The test contains thirteen logic and reasoning questions with a two-minute time limit and was designed as a measure of fluid intelligence. Each respondent took the test up to four times, and we use the mean of the standardized scores, which was then standardized (and residualized, as described in the UKB row of Table S1.4).”

(COGENT Consortium) “For each of 35 component studies in the meta-analysis, the phenotype used was the first unrotated principal component of performance on at least 3 neuropsychological tests (or at least two IQ-test subscales). In general, the test variable used measured overall accuracy or total number of correct responses. Across the individual studies, the first PC explained an average of 41% of the variance in test performance. The average Cronbach's alpha (a measure of internal consistency between test items) was 0.70 across component studies.”

6. [Line 185] Given that the authors included two exposures, the Bonferroni-corrected significance threshold should be $0.05/(2*10)$.

-We thank Reviewer 1 for noting this. We have corrected the Bonferroni-corrected significance threshold to be 2.5×10^{-3} in Methods:

“Associations with a p-value $< 2.5 \times 10^{-3}$ (Bonferroni-corrected significance threshold accounting for two exposures and ten outcomes) were considered significant.”

Our conclusions remain consistent.

7. [Line 192-195] Could the authors clarify their definition of high and moderate risk of horizontal pleiotropy?

-We have clarified in Methods:

“We queried each genetic instrument in the Open Targets Genetics database^{9,10} (<https://genetics.opentargets.org/>, accessed December 15, 2023), which includes existing GWASs in the GWAS Catalog as well as GWASs conducted using the UK Biobank or FinnGen resources¹¹⁻¹⁴. After excluding associations with vitamin B12 levels, folate levels, and pernicious anemia, we defined genetic instruments that are associated with one to five phenotypes (p-value <5.0x10⁻⁸) to be subject to a moderate risk of horizontal pleiotropy, and those associated with more than five phenotypes to be subject to a high risk of horizontal pleiotropy. This ad hoc classification was adopted to enable multi-layered sensitivity analyses, although it could not distinguish between horizontal and vertical pleiotropy. All genotype-phenotype associations involving the genetic instruments used in this study are summarized in **Supplementary Table S2.**”

As mentioned in the text, this classification was introduced to enable multi-layered sensitivity analyses, as existing horizontal pleiotropy-robust MR methods (e.g., MR-Egger) rely on additional assumptions (e.g., the InSIDE assumption) that may not hold and cannot be directly verified in real-world data. For our primary analyses, we used all available instruments regardless of whether they demonstrated pleiotropy.

8. [Supplementary Table S4] Did the authors use only pernicious anemia or all types of anemia as positive control outcomes? Please explain the selection of positive control outcomes in the method section.

-We appreciate this comment. We only considered pernicious anemia as the positive control outcome, as other types of anemia involve distinct biological mechanisms. We have clarified in Methods:

“As a positive control, we assessed whether genetically predicted vitamin B12 and folate levels were associated with pernicious anemia.”

“We queried each genetic instrument for its association with pernicious anemia, based on the International Classification of Diseases version 10 (ICD-10) code of D51 or self-reported physician-made diagnoses. Additionally, we investigated associations with other types of anemia for comparison.”

9. [Supplementary Tables S5-6] Could the authors report the true effect size for 80% power rather than several columns of power with differing true effects? Please also make relevant changes in the Methods and Results sections.

-We thank Reviewer 1 for this suggestion. We have revised to report the true effect sizes for 80% power with a type I error rate of 5% and 0.25%, respectively. We have also revised in Methods and Results:

“For each outcome, we calculated the required true effect size associated with a one standard deviation change in vitamin B12 levels to achieve 80% power, with a type I error rate (i.e., significance level) of 5% and 0.25% (Bonferroni-corrected significance threshold accounting for two exposures and ten outcomes), respectively.”

“As a result, with a type I error rate of 0.25% (Bonferroni-corrected significance threshold), MR analyses with depression as the outcome could achieve 80% power when the true OR per one standard deviation change in vitamin B12 levels was ≥ 1.03 (**Supplementary Table S5**). For educational attainment and cognitive performance, MR analyses could achieve 80% power when the true effect sizes were ≥ 0.020 and ≥ 0.034 per one standard deviation change in vitamin B12 levels, respectively (**Supplementary Table S5**).”

10. [Discussion] Could the authors compare their findings with previous evidence from trials and observational studies? Could the authors provide more biological explanations of their findings?

-We thank Reviewer 1 for this suggestion. One important limitation of our study is that we could not directly consider orthogonal evidence due to insufficient data availability. For example, we do not have access to cohort studies where vitamin B12 levels and the outcomes of interest are simultaneously available. We have included in Introduction mentioning known biological functions of vitamin B12 as well as existing trials and observational studies:

“Together with folate, vitamin B12 has multiple essential roles in various biological processes, such as DNA synthesis and methylation, maturation of erythrocytes, as well as fatty acid and amino acid metabolism^{15–17}. Vitamin B12 also contributes to the maintenance of the nervous system via the synthesis of myelin and neuronal regeneration^{18,19}.”

“Untreated vitamin B12 deficiency can lead to pernicious anemia and can result in neurological manifestations, such as peripheral neuropathy, ataxia, various psychiatric disorders, as well as cognitive impairment^{20–22}. Amongst individuals with depression or schizophrenia, randomized placebo-controlled trials have shown that folate plus vitamin B12 supplementation may preserve cognitive functions and delay the disease progression^{23–25}. However, randomized placebo-controlled trials have not elucidated whether increasing vitamin B12 levels through supplementation can help to prevent the onset of psychiatric disorders and cognitive impairment in the general population²⁶, where the majority does not have clinical vitamin B12 deficiency. Although a longitudinal study indicated that vitamin B12 levels are negatively associated with incident depressive symptoms in older adults²⁷, these observational associations may be subject to confounding factors that are difficult to fully account for, such as socioeconomic status, lifestyle factors, and comorbidities.”

Notably, our study yielded null results, which require careful interpretation. First, MR relies on the implicit assumption that long-term genetic effects are equivalent to drug effects, which may not always hold. Second, these null results do not preclude the possibility of short-term or high-dose effects of vitamin B12 in specific subgroups or clinical contexts. Therefore, our findings should not be interpreted as evidence against the established biological functions of vitamin B12.

Rather, they reflect an assessment of the potential impact of modulating serum vitamin B12 levels in the general population. We have discussed these points in Discussion:

“Our study has a clear and important implication that, in the general population, vitamin B12 supplementation is unlikely to meaningfully reduce the risks of the investigated psychiatric disorders or significantly improve educational attainment or cognitive performance.”

“Importantly, our findings should not be interpreted as refuting any known biological functions of vitamin B12, particularly in individuals with deficiency or specific medical conditions where supplementation is clinically indicated. Multiple lines of evidence are still needed to ascertain the potential impact of vitamin B12 or folate plus vitamin B12 supplementation on various outcomes in individuals with vitamin B12 deficiency.”

“Some limitations of our study should be noted. First, our MR analyses were not triangulated with observational estimates due to insufficient data availability.”

“Last, our results do not provide information on whether serum vitamin B12 levels may influence the progression of psychiatric disorders or cognitive impairment, as the genetic architecture underlying disease progression may differ from that underlying disease pathogenesis. We anticipate that future investigations will more comprehensively illuminate the role of vitamin B12 in various psychiatric disorders and cognitive impairment in diverse populations.”

11. [Line 297-299] The authors selected genetic variants significantly associated with serum vitamin B12 and folate levels with a p-value $< 2.2 \times 10^{-9}$, which is a statistical approach. Could the authors explain more about their crucial roles in the absorption, transport, or enzymatic reactions of vitamin B12 and why it is relevant to unmeasured confounding and reverse causation?

-We appreciate this comment. First of all, we apologize for any confusion in our previous narrative. The biology underlying the genetic instruments is unrelated to unmeasured confounding and reverse causation. The reduced risk of confounding and reverse causation is a feature of MR analyses (though, of course, with other caveats discussed in the manuscript). We have clarified in Discussion that we meant the selected instruments are strong, as indicated by their F-statistics:

“First, we employed genetic instruments that were protein-altering variants affecting crucial genes in the absorption, transport, or enzymatic reactions of vitamin B12 for MR analyses, which effectively guarded against weak instrument bias (**Supplementary Table S1**).”

Furthermore, all of these genetic instruments are missense or stop-gain variants that can be explicitly mapped to protein-coding genes. The functions of these genes are summarized in **Supplementary Table S1**:

MAPPED GENE	FUNCTIONAL IMPACT ON MAPPED GENE	NCBI GENE SUMMARY
MMACHC	Missense variant (p.Arg206Gln)	The exact function of the protein encoded by this gene is not known, however, its C-terminal region shows similarity to TonB, a bacterial protein involved in energy transduction for cobalamin (vitamin B12) uptake. Hence, it is postulated that this protein may have a role in the binding and intracellular trafficking of cobalamin. Mutations in this gene are associated with methylmalonic aciduria and homocystinuria type cblC. [provided by RefSeq, Oct 2009]
MMAA	Missense variant (p.Gln363His)	The protein encoded by this gene is involved in the translocation of cobalamin into the mitochondrion, where it is used in the final steps of adenosylcobalamin synthesis. Adenosylcobalamin is a coenzyme required for the activity of methylmalonyl-CoA mutase. Defects in this gene are a cause of methylmalonic aciduria. [provided by RefSeq, Jul 2008]
MUT	Missense variant (p.Arg532His)	This gene encodes the mitochondrial enzyme methylmalonyl Coenzyme A mutase. In humans, the product of this gene is a vitamin B12-dependent enzyme which catalyzes the isomerization of methylmalonyl-CoA to succinyl-CoA, while in other species this enzyme may have different functions. Mutations in this gene may lead to various types of methylmalonic aciduria. [provided by RefSeq, Jul 2008]
CUBN	Missense variant (p.Phe253Ser)	Cubilin (CUBN) acts as a receptor for intrinsic factor-vitamin B12 complexes. The role of receptor is supported by the presence of 27 CUB domains. Cubulin is located within the epithelium of intestine and kidney. Mutations in CUBN may play a role in autosomal recessive megaloblastic anemia. [provided by RefSeq, Jul 2008]
TCN1	Missense variant (p.Asp301Tyr)	This gene encodes a member of the vitamin B12-binding protein family. This family of proteins, alternatively referred to as R binders, is expressed in various tissues and secretions. This protein is a major constituent of secondary granules in neutrophils and facilitates the transport of cobalamin into cells. [provided by RefSeq, Jul 2008]
CLYBL	Stop-gain variant (resulting in the substitution of arginine at position 259 with a stop codon)	Enables (S)-citramalyl-CoA lyase activity; magnesium ion binding activity; and malate synthase activity. Involved in protein homotrimerization and regulation of cobalamin metabolic process. Predicted to be located in mitochondrion. Predicted to be integral component of membrane. [provided by Alliance of Genome Resources, Apr 2022]
ABCD4	Missense variant (p.Glu368Lys)	The protein encoded by this gene is a member of the superfamily of ATP-binding cassette (ABC) transporters. ABC proteins transport various molecules across extra- and intra-cellular membranes. ABC genes are divided into seven distinct subfamilies (ABC1, MDR/TAP, MRP, ALD, OABP, GCN20, White). This protein is a member of the ALD subfamily, which is involved in peroxisomal import of fatty acids and/or fatty acyl-CoAs in the organelle. All known peroxisomal ABC transporters are half transporters which require a partner half transporter molecule to form a functional homodimeric or heterodimeric transporter. The function of this peroxisomal membrane protein is unknown. However, it is speculated that it may function as a heterodimer for another peroxisomal ABC transporter and, therefore, may modify the adrenoleukodystrophy phenotype. It may also play a role in the process of peroxisome biogenesis. Alternative splicing results in several protein-coding and non-protein-coding variants. [provided by RefSeq, Jul 2017]

FUT6	Missense variant (p.Pro124Ser)	The protein encoded by this gene is a Golgi stack membrane protein that is involved in the creation of sialyl-Lewis X, an E-selectin ligand. Mutations in this gene are a cause of fucosyltransferase-6 deficiency. Two transcript variants encoding the same protein have been found for this gene. [provided by RefSeq, Jul 2008]
CD320	Missense variant (p.Gly220Arg)	This gene encodes the transcobalamin receptor that is expressed at the cell surface. It mediates the cellular uptake of transcobalamin bound cobalamin (vitamin B12), and is involved in B-cell proliferation and immunoglobulin secretion. Mutations in this gene are associated with methylmalonic aciduria. Alternatively spliced transcript variants encoding different isoforms have been found for this gene.[provided by RefSeq, Jan 2011]
FUT2	Missense variant (p.Gly258Ser)	This gene is one of two encoding the galactoside 2-L-fucosyltransferase enzyme. The encoded protein is important for the final step in the soluble ABO blood group antigen synthesis pathway. It is also involved in cell-cell interaction, cell surface expression, and cell proliferation. Mutations in this gene are a cause of the H-Bombay blood group where red blood cells lack the H antigen. [provided by RefSeq, May 2022]
TCN2	Missense variant (p.Leu376Ser)	This gene encodes a member of the vitamin B12-binding protein family. This family of proteins, alternatively referred to as R binders, is expressed in various tissues and secretions. This plasma protein binds cobalamin and mediates the transport of cobalamin into cells. This protein and other mammalian cobalamin-binding proteins, such as transcobalamin I and gastric intrinsic factor, may have evolved by duplication of a common ancestral gene. Alternative splicing results in multiple transcript variants.[provided by RefSeq, May 2010]

12. [Line 307] Could the authors explain why the MR estimates for folate level are more prone to horizontal pleiotropy?

-The two instruments for serum folate levels have been associated with multiple traits, as mentioned in Results:

“In addition, two independent genetic variants, rs1801133 and rs652197, were significantly associated with serum folate levels (p-value <2.2x10⁻⁹; **Supplementary Table S1**). However, rs1801133, a missense variant of *MTHFR*, demonstrated a high risk of horizontal pleiotropy, with known associations with N-terminal prohormone brain natriuretic peptide levels, blood pressure, and multiple blood cell characteristics (**Supplementary Table S2**). Meanwhile, rs652197, an intronic variant of *FOLR3*, has been associated with serum 25-hydroxyvitamin D levels in previous studies (**Supplementary Table S2**).”

Although these traits themselves may not have a true causal effect on the outcomes of interest, the MR estimates cannot be interpreted as the effects of serum folate levels as long as such pleiotropy is horizontal.

Furthermore, since there were only two instruments for serum folate levels, existing horizontal pleiotropy-robust MR methods could not be applied, which all require at least three instruments. Therefore, our manuscript focused on serum vitamin B12 levels rather than folate levels, as explained in Results and Discussion:

“Given the risk of horizontal pleiotropy in the instruments for folate levels and the inapplicability of MR sensitivity analysis methods, subsequent analyses and interpretation of findings did not focus on folate levels.”

“Nevertheless, MR estimates for folate levels were more prone to bias due to horizontal pleiotropy, thus interpreting the results involving folate levels requires extra caution.”

Reviewer #2 (Remarks to the Author):

Lu and Paterson conducted an MR study to explore the association between serum vitamin B12 levels and psychiatric disorders and cognitive impairment. They found no association between genetically predicted serum vitamin B12 levels and either of the eight psychiatric disorders, educational attainment or cognitive performance. Even though the analysis was well performed, these null associations were reported in previous studies, like some studies cited by the authors, a recent PheWAS on B vitamins (pmid: 36811473), and this MR study (pmid: 38999734). These findings provide limited novelty.

-We thank Reviewer 2 for reviewing our work and for providing their comments. However, we respectfully disagree with the comment regarding limited novelty.

First, our study has substantially increased power to detect true effects, if any, compared to these two studies mentioned by Reviewer 2, since we leveraged some of the largest GWAS meta-analyses of psychiatric disorders. In contrast, Dib et al.²⁸ only focused on outcomes in the UK Biobank while Hu et al.²⁹ only focused on outcomes in the FinnGen. For example, Dib et al. included 1,393 cases of bipolar disorder, 21,502 cases of depression, and 788 cases of schizophrenia, while our analyses included a total of 41,917 cases of bipolar disorder, 550,355 cases of depression, and 69,369 cases of schizophrenia. We have mentioned in Introduction:

“Recently, a phenome-wide study based on the UK Biobank confirmed the association between increased vitamin B12 levels and reduced odds of pernicious anemia²⁸. However, it remains unclear whether the genetically predicted vitamin B12 levels are associated with psychiatric disorders and cognitive impairment, due to limited statistical power in previous studies.”

Furthermore, we have major reservations about the study conducted by Hu et al.²⁹, in which the genetic instruments for “vitamin B12” were obtained from a GWAS conducted in the UK Biobank. However, the UK Biobank did not actually measure serum vitamin B12 levels. Instead, the GWAS used data field 100013 (<https://gwas.mrcieu.ac.uk/datasets/ukb-b-19524/>; <https://biobank.ndph.ox.ac.uk/showcase/field.cgi?id=100013>), which represents “Estimated intake, based on food and beverage consumption yesterday” (based on questionnaire data). Genetic associations with estimated intake of vitamin B12, which can be influenced by numerous environmental, lifestyle, and socioeconomic factors but not likely by biological factors underlying the metabolism of vitamin B12, substantially differed from those with measured serum vitamin B12 levels³⁰. Notably, none of the “instruments” (which did not reach genome-wide significance, as the authors used a relaxed threshold of p-value $<5 \times 10^{-6}$) overlapped with previously identified genomic risk loci for serum vitamin B12 levels³⁰. In short, this study, as far as we are concerned, is not testing for the potential effects of serum vitamin B12 levels.

Second, our study addresses potential confounding effects that may bias MR analyses involving psychosocial traits by leveraging within-sibship GWASs for additional sensitivity analyses—an issue that has only recently (re-)gained attention in the field. We have mentioned in Introduction:

“In particular, although the randomization of genetic variants at conception breaks associations with many conventional confounders, MR analyses, particularly those involving psychosocial traits, may still be biased by confounding due to uncontrolled demographic effects and indirect genetic effects¹. These biases can be mitigated by using data from family-based studies¹⁻³.”

Third, in addition to applying horizontal pleiotropy-robust MR methods, our study conducted multi-layered sensitivity analyses by excluding pleiotropic instruments. These complementary approaches are important, as all horizontal pleiotropy-robust MR methods come with additional assumptions that may not always be satisfied in real data.

In summary, while our study reached the same conclusion that serum vitamin B12 levels may not have a substantial protective effect against psychiatric disorders or cognitive impairment in the general population, our findings offer significantly improved credibility and robustness compared to existing work.

Reviewer #3 (Remarks to the Author):

Lu et al. explored associations of serum vitamin B12 with psychiatric disorders and cognitive impairment using Mendelian randomization (MR). My specific comments are below:

-We appreciate comments and suggestions from Reviewer 3.

1. Line 74, in terms of the 2nd MR assumption, please specify the confounders.

-We have specified the second assumption and provided more details in Introduction:

“the genetic instrument-outcome association should not be confounded”

“In particular, although the randomization of genetic variants at conception breaks associations with many conventional confounders, MR analyses, particularly those involving psychosocial traits, may still be biased by confounding due to uncontrolled demographic effects and indirect genetic effects¹. These biases can be mitigated by using data from family-based studies¹⁻³.”

2. Please add suitable sensitivity analysis for folate, as it only has two genetic instruments.

-We appreciate this comment. However, we recognize important limitations of the analyses involving serum folate levels. First, the instruments for serum folate levels demonstrated pleiotropy, as mentioned in Results:

“However, rs1801133, a missense variant of *MTHFR*, demonstrated a high risk of horizontal pleiotropy, with known associations with N-terminal prohormone brain natriuretic peptide levels, blood pressure, and multiple blood cell characteristics (**Supplementary Table S2**). Meanwhile, rs652197, an intronic variant of *FOLR3*, has been associated with serum 25-hydroxyvitamin D levels in previous studies (**Supplementary Table S2**).”

Second, since there were only two instruments for serum folate levels, existing horizontal pleiotropy-robust MR methods could not be applied, which all require at least three instruments. Therefore, our manuscript focused on serum vitamin B12 levels rather than folate levels, as explained in Results and Discussion:

“Given the risk of horizontal pleiotropy in the instruments for folate levels and the inapplicability of MR sensitivity analysis methods, subsequent analyses and interpretation of findings did not focus on folate levels.”

“Nevertheless, MR estimates for folate levels were more prone to bias due to horizontal pleiotropy, thus interpreting the results involving folate levels requires extra caution.”

3. In line 180 – 189, please briefly summarize how each MR sensitivity analysis deals with potential violations of MR assumption(s).

-We have included more information about these methods:

“We performed MR to assess the association between the genetically predicted vitamin B12 levels and each type of anemia using the weighted median method due to the presence of pleiotropy in several genetic instruments (**Supplementary Table S2**). The weighted median method estimates the effect size of the exposure by calculating a weighted median of the Wald ratio estimates from individual genetic instruments, with weights corresponding to the inverse variance of each estimate. This method can provide robust estimates with up to 50% of the information coming from invalid genetic instruments⁴. We additionally performed MR using the inverse variance weighted regression, penalized weighted median, simple mode, weighted mode, and MR-Egger methods. Specifically, the inverse variance weighted regression method assumes that all instruments are valid⁵. The penalized weighted median method, similar to the weighted median method, assumes that the majority of the instruments are valid⁴. The simple mode and weighted mode methods assume that the plurality of the instruments are valid³¹. The MR-Egger method assumes that the strength of the instruments is independent of their direct effects on the outcome³². A nominally significant MR-Egger intercept (p-value <0.05) was considered evidence of directional pleiotropy³².”

The statistical details of these methods can be found in the referenced literature.

4. Please also explain why weighted median approach was used as the main results.

-We appreciate this suggestion. We have added in Methods:

“We performed MR to assess the association between the genetically predicted vitamin B12 levels and each type of anemia using the weighted median method due to the presence of pleiotropy in several genetic instruments (**Supplementary Table S2**). The weighted median method estimates the effect size of the exposure by calculating a weighted median of the Wald ratio estimates from individual genetic instruments, with weights corresponding to the inverse variance of each estimate. This method can provide robust estimates with up to 50% of the information coming from invalid genetic instruments⁴.”

5. GWAS of depression (i.e. reference 38) is out-of-date. Please consider the most recent one.

-We appreciate this suggestion. We have now used the most recent GWAS for depression published in Jan 2025 that included 550,355 cases and 2,071,918 controls³³. All results have been updated accordingly and our findings remain consistent.

6. Lines 192 – 193, please explain how to discriminate horizontal pleiotropy from vertical pleiotropy.

-We thank Reviewer 3 for this question. In fact, we cannot distinguish between horizontal and vertical pleiotropy, as this would require determining whether a trait fully mediates the genetic effects on vitamin B12 or whether vitamin B12 fully mediates the genetic effects on the trait—

both of which remain elusive in this study. Here, we simply used the number of pleiotropic associations as a proxy for the risk of horizontal pleiotropy, which, of course, has limitations. However, we introduced this classification of “high” and “moderate” risk of horizontal pleiotropy to enable multi-layered sensitivity analyses, as existing horizontal pleiotropy-robust MR methods (e.g., MR-Egger) rely on additional assumptions (e.g., the InSIDE assumption) that may not hold and cannot be directly verified in real-world data. For our primary analyses, we used all available instruments regardless of whether they demonstrated pleiotropy. We have clarified in Methods:

“After excluding associations with vitamin B12 levels, folate levels, and pernicious anemia, we defined genetic instruments that are associated with one to five phenotypes (p -value $< 5.0 \times 10^{-8}$) to be subject to a moderate risk of horizontal pleiotropy, and those associated with more than five phenotypes to be subject to a high risk of horizontal pleiotropy. This ad hoc classification was adopted to enable multi-layered sensitivity analyses, although it could not distinguish between horizontal and vertical pleiotropy. All genotype-phenotype associations involving the genetic instruments used in this study are summarized in **Supplementary Table S2.**”

7. This study is a two-sample MR. Please comment on whether there is a sample overlap. If there is no sample overlap, then it would be a strength.

-We appreciate this suggestion. We have added in Methods:

“The samples included in the exposure GWASs partially overlapped with those in the outcome GWASs for bipolar disorder, depression, schizophrenia, and educational attainment. However, the expected bias towards observational estimates were minimal given the large F-statistics of the instruments³⁴ (**Supplementary Table S1**). For the other outcome GWASs, there was no known sample overlap.”

Importantly, although the GWAS for serum vitamin B12 levels overlapped with the GWASs for bipolar disorder (total N = 413,466), depression (total N = 2,622,273), schizophrenia (total N = 306,011), and educational attainment (total N = 765,283), the maximum number of overlapping samples is 37,283 (sample size of a sub-cohort in the deCODE study)³⁰. This corresponds to a maximum sample overlap of <10%. The exact proportion of sample overlap is difficult to determine, as we lack information on whether the same 37,283 deCODE participants were included in the GWASs for the outcomes and on the proportion of cases and controls among them. Nevertheless, for each instrument, we provide the expected proportion of bias towards observational estimate in the case of full (100%) sample overlap (bias = $1 / F$ -statistics³⁴) in Supplementary Table S1, suggesting that sample overlap has a limited impact on this study:

SNP	F-STATISTIC	EXPECTED PROPORTION OF BIAS TOWARDS (CONFOUNDED) OBSERVATIONAL ESTIMATE WITH FULL SAMPLE OVERLAP*
RS371753672	3582.30	0.03%
RS2270655	673.36	0.15%
RS1141321	316.78	0.32%
RS1801222	316.78	0.32%
RS34324219	495.80	0.20%
RS41281112	673.36	0.15%
RS3742801	316.78	0.32%
RS778805	316.78	0.32%
RS2336573	849.45	0.12%
RS602662	316.78	0.32%
RS1131603	673.36	0.15%

*In this study, the proportion of sample overlap is less than 10%

8. Please add a DAG to explain MR based on within-sibship GWAS in Figure 1.

-We have added an illustrative panel in **Supplementary Figure S1** (previously Figure 1, changed to a supplementary figure based on comments from Reviewer 4).

Supplementary Figure S1. Overview of study design. Summary statistics of large-scale genome-wide association studies were obtained for serum vitamin B12 and folate levels, eight psychiatric disorders, educational attainment, and cognitive performance. Mendelian randomization and multiple sensitivity analyses were performed to estimate the potential effects of vitamin B12 and folate levels. Within-sibship studies reduce confounding in population-based genome-wide association studies by controlling for demographic effects and indirect genetic effects.

9. The numbers of cases for anxiety disorders and obsessive-compulsive disorder are insufficient. Please consider meta-analyzing the GWAS currently used with other publicly available GWAS (e.g. from FinnGen) of the same outcomes.

-We appreciate this suggestion and have identified more recent GWASs for these two disease outcomes. For anxiety disorders, we have used a more recent meta-analysis with an effective sample size of 529,764³⁵. Moreover, we have replaced the original case-control GWAS for OCD with a meta-analysis of GWASs for obsessive-compulsive symptoms (OCS; N = 33,943)³⁶, which were measured using standardized obsessive compulsive item scores. The OCS GWAS was better powered due to the use of a continuous outcome³⁶. We refrained from conducting new meta-analyses because psychiatric disorders can be highly heterogeneous, and their definitions can vary across cohorts. Addressing these complexities is beyond the scope of this study.

All results have been updated accordingly and our findings remain consistent. The updated GWASs for depression, anxiety disorders, and OCS have substantially enhanced the credibility and robustness of our work.

Reviewer #4 (Remarks to the Author):

It is my pleasure to review this manuscript. The authors applied comprehensive Mendelian randomization methods to explore the effect of vitamin B12 on eight psychiatric disorders and found no effect of vitamin B12 on any of these disorders. It is a well conducted study. Although it suggests null results, it still worthy to be addressed in the nutritional epidemiology and has an insight in clinical practise. On top of these, there are some concerns from my end, which might be additional before going to an official publication:

-We appreciate comments and suggestions from Reviewer 4.

Major:

- The exome-sequencing data focus on protein-coding regions within the genome. It represents only about 1-2% of the entire genome but contain the majority of disease-causing variants. For the positive control of pernicious anemia, can the author clarify why only the exome-sequence data is applied? Are there any general GWAS data available? If so, why please clarify why the general GWAS data is not applied. I imagine the exome-sequencing data for the other 8 outcomes are not available?

-We thank Reviewer 4 for this question. We used the exome-sequencing data because all of the instruments are missense or stop-gain variants in the coding regions, for which the exome-sequencing data provide higher accuracy and direct detection of variants without relying on imputation. We have clarified in Methods:

“Exome-sequencing data were prioritized over genotyping and imputation data because the instruments used for MR analyses are located in coding regions (**Supplementary Table S1**), where exome sequencing provides higher accuracy and direct detection of variants without relying on imputation⁶.”

For the analyses of psychiatric disorders, educational attainment, and cognitive performance, although these outcomes are available in the UK Biobank, we utilized large-scale meta-analyses of general GWASs across multiple cohorts (**Table 1**), which provided significantly larger sample sizes—and therefore greater statistical power—compared to UK Biobank-only whole-exome sequencing data.

To ensure the validity of our results, we repeated the MR analyses using GWAS summary statistics from the Pan-UK Biobank⁷, which were derived from genotyping and imputation data. The positive control associations with pernicious anemia remained significant and showed consistent effect magnitudes compared to those obtained using whole-exome sequencing data (**Additional Figures 1 and 2**). For conciseness, these additional analyses are not included in the main manuscript.

Additional Figure 1. Associations between genetically predicted vitamin B12 levels and various types of anemia based on Pan-UK Biobank GWASs.

Additional Figure 2. Associations between genetically predicted vitamin B12 levels and various types of anemia based on Pan-UK Biobank GWASs.

- Please clarify what are missense or stop-gain variants. It is not friendly to the audience who are not familiar with this advance genetic terminology. I believe this is one of the reasons why you apply exome-sequencing GWAS. Besides, any general GWAS for vitamin B12 is available? Though the genetic variants obtained from general GWAS are likely subject to the violations of the IV assumptions. But I think this can be briefly discussed.

-Indeed, we used the exome-sequencing data because all of the instruments are missense or stop-gain variants in the coding regions, for which the exome-sequencing data provide higher accuracy and direct detection of variants without relying on imputation. We have added explanations for missense and stop-gain variants in Results:

“All 11 instruments are either missense variants, which result in a different amino acid in the protein, or stop-gain variants, which lead to truncated proteins, affecting protein-coding genes. (Supplementary Table S1).”

- Although the observational design is open to confounding, is it available to conduct an observational design in this study? Triangulation the estimates would be appreciated.

-We appreciate this suggestion. Unfortunately, we do not have access to study cohorts (e.g. the deCODE study, which does not allow access to individual-level data) that have simultaneously measured serum vitamin B12 levels and the outcomes of interest. We have included in Introduction mentioning existing trials and observational studies:

“Untreated vitamin B12 deficiency can lead to pernicious anemia and can result in neurological manifestations, such as peripheral neuropathy, ataxia, various psychiatric disorders, as well as cognitive impairment^{20–22}. Amongst individuals with depression or schizophrenia, randomized placebo-controlled trials have shown that folate plus vitamin B12 supplementation may preserve cognitive functions and delay the disease progression^{23–25}. However, randomized placebo-controlled trials have not elucidated whether increasing vitamin B12 levels through supplementation can help to prevent the onset of psychiatric disorders and cognitive impairment in the general population²⁶, where the majority does not have clinical vitamin B12 deficiency. Although a longitudinal study indicated that vitamin B12 levels are negatively associated with incident depressive symptoms in older adults²⁷, these observational associations may be subject to confounding factors that are difficult to fully account for, such as socioeconomic status, lifestyle factors, and comorbidities.”

We have also discussed this important limitation in Discussion:

“However, it is noteworthy that none of the GWASs utilized in this study were based on populations selected for vitamin B12 deficiency. Importantly, our findings should not be interpreted as refuting any known biological functions of vitamin B12, particularly in individuals with deficiency or specific medical conditions where supplementation is clinically indicated. Multiple lines of evidence are still needed to ascertain the potential impact of vitamin B12 or folate plus vitamin B12 supplementation on various outcomes in individuals with vitamin B12 deficiency.”

“Some limitations of our study should be noted. First, our MR analyses were not triangulated with observational estimates due to insufficient data availability.”

“We anticipate that future investigations will more comprehensively illuminate the role of vitamin B12 in various psychiatric disorders and cognitive impairment in diverse populations.”

- I agree that it is important to account for the potential pleiotropy in an MR design. But the standard of how to define a “moderate” risk of pleiotropy is not clear. For example, “Missense variants in MUT, FUT6, CD320, and CUBN were subject to a moderate risk of horizontal pleiotropy

due to their associations with other phenotypes, such as liver function biomarkers, lipid levels, and height.” It is not convincing to me that these phenotypes are causally associated with psychiatric disorders. I appreciate the way of excluding the potential pleiotropy SNPs to account for the violation of the exclusion restriction assumption. But the exclusion criteria should be clear and with scientific evidence (e.g., relatively clear evidence of being a causal phenotype of the outcome of interest).

-We thank Reviewer 4 for this comment. We introduced this classification of “high” and “moderate” risk of horizontal pleiotropy to enable multi-layered sensitivity analyses, as existing horizontal pleiotropy-robust MR methods (e.g., MR-Egger) rely on additional assumptions (e.g., the InSIDE assumption) that may not hold and cannot be directly verified in real-world data. Importantly, for our primary analyses, we used all available instruments regardless of whether they demonstrated pleiotropy. We have clarified in Methods:

“After excluding associations with vitamin B12 levels, folate levels, and pernicious anemia, we defined genetic instruments that are associated with one to five phenotypes (p -value $< 5.0 \times 10^{-8}$) to be subject to a moderate risk of horizontal pleiotropy, and those associated with more than five phenotypes to be subject to a high risk of horizontal pleiotropy. This ad hoc classification was adopted to enable multi-layered sensitivity analyses, although it could not distinguish between horizontal and vertical pleiotropy. All genotype-phenotype associations involving the genetic instruments used in this study are summarized in **Supplementary Table S2.**”

We agree with Reviewer 4 that if a trait does not have a causal effect on the outcome of interest, the association between the instrument and the trait does not violate the exclusion restriction assumption. However, examining whether this trait has a causal effect on the outcome is a separate question that requires investigation potentially including MR and other lines of evidence, which is beyond the scope of this study.

With these considerations, our multi-layered analyses can be interpreted as follows:

- (1) Primary analysis: The most relaxed criterion for instrument inclusion;
- (2) Sensitivity analysis excluding high-risk instruments: A more stringent criterion for instrument inclusion, with a risk of losing valid instruments that are not actually subject to horizontal pleiotropy;
- (3) Sensitivity analysis excluding high- and moderate-risk instruments: The most stringent criterion for instrument inclusion, with a higher risk of losing valid instruments that are not actually subject to horizontal pleiotropy.

The consistency of results across all analyses, which remained null, further strengthens the credibility and robustness of our findings.

• If I understand correctly, Figure 2 is missing. Figure 3 in the manuscript is Figure 2 and Figure 4 in the manuscript is Figure 3.

-We thank Reviewer 4 for noting this. We have corrected the labelling of the figures.

Minor:

- It would be more straightforward that unit of presenting the point estimate in regular format (e.g., a 0.008 in SD unit, 95% CI: -0.010 to 0.025, p-value = 0.39) and make it consistent with the number of decimal place.

-We appreciate this suggestion. We have revised the statistics to be reported with two significant digits throughout the manuscript.

- Line 126, it is not clear to me what are these five other phenotypes.

-We apologize for any confusion in our previous narrative. We meant to state that if an instrument is associated with >5 phenotypes, it would be considered to be subject to a high risk of horizontal pleiotropy, and that if an instrument is associated with 1, 2, 3, 4, or 5 phenotypes, it would be considered to be subject to a moderate risk of horizontal pleiotropy. We introduced this classification of “high” and “moderate” risk of horizontal pleiotropy to enable multi-layered sensitivity analyses, as existing horizontal pleiotropy-robust MR methods (e.g., MR-Egger) rely on additional assumptions (e.g., the InSIDE assumption) that may not hold and cannot be directly verified in real-world data. For our primary analyses, we used all available instruments regardless of whether they demonstrated pleiotropy. We have clarified in Methods:

“After excluding associations with vitamin B12 levels, folate levels, and pernicious anemia, we defined genetic instruments that are associated with one to five phenotypes (p-value <5.0x10⁻⁸) to be subject to a moderate risk of horizontal pleiotropy, and those associated with more than five phenotypes to be subject to a high risk of horizontal pleiotropy. This ad hoc classification was adopted to enable multi-layered sensitivity analyses, although it could not distinguish between horizontal and vertical pleiotropy. All genotype-phenotype associations involving the genetic instruments used in this study are summarized in **Supplementary Table S2.**”

- Line 173, I imagine the SD unit in the power calculation is based on 5% type I error and 80% power.

-We have now clarified in Methods:

“For each outcome, we calculated the required true effect size associated with a one standard deviation change in vitamin B12 levels to achieve 80% power, with a type I error rate (i.e., significance level) of 5% and 0.25% (Bonferroni-corrected significance threshold accounting for two exposures and ten outcomes), respectively.”

- Please forgive this is a subjective comment, I don't think Figure 1 is not very helpful (e.g., not clear in terms of study design and/or flow) for the audiences to have a quick understanding of what this study is about and how it is conducted.

-We thank Reviewer 4 for this comment. We intended to use this figure to showcase data resources as well as elements of the analyses. We have now moved it to the supplementary materials, as **Supplementary Figure S1**.

Supplementary Figure S1. Overview of study design. Summary statistics of large-scale genome-wide association studies were obtained for serum vitamin B12 and folate levels, eight psychiatric disorders, educational attainment, and cognitive performance. Mendelian randomization and multiple sensitivity analyses were performed to estimate the potential effects of vitamin B12 and folate levels. Within-sibship studies reduce confounding in population-based genome-wide association studies by controlling for demographic effects and indirect genetic effects.

- Line 231, “rs1801133, a missense variant of MTHFR, demonstrated a high risk of horizontal pleiotropy”, what are the phenotypes of the high risk of horizontal pleiotropy?

-We have added in Results:

“However, rs1801133, a missense variant of *MTHFR*, demonstrated a high risk of horizontal pleiotropy, with known associations with N-terminal prohormone brain natriuretic peptide levels, blood pressure, and multiple blood cell characteristics (**Supplementary Table S2**).”

- I appreciate that the author has done lots of work on the power calculation. For clinical / public health perspective, I would suggest the author assign/set up a “clinical effect size” (like a benchmark), which can be based on the clinical and public health background/information (e.g., combing the prevalence of the disease outcome and if an effective medical treatment is available, an ideal effect size e.g., OR = 1.01 would be, for example, 10% cases of the disease can be

prevented in the population. Therefore, we consider this a reference to discuss whether the MR estimate we obtained has enough power). In this case, it provides an intuitive understanding of where a practical effect size would be.

-We appreciate this comment. However, translating odds ratios to absolute risk requires knowledge of disease prevalence. For instance, if the baseline disease prevalence is 1.00%, an OR of 2 corresponds to an increased prevalence of 1.98%, whereas for a baseline prevalence of 10.00%, the same OR corresponds to an increased prevalence of 18.18%.

In this study, determining the baseline prevalence of psychiatric disorders is challenging due to their high heterogeneity, even within populations of the same genetic ancestry. Given that our findings are null, we have opted not to discuss the clinical relevance of effect sizes.

- Is it possible to access how many pmol per litre as indicated by 1 SD unit? Also, what is the distribution of Vitamin B12 level in the population? If the Vitamin B12 is skewed distributed, a 1SD change is not ideal to interpret the results, which can be acknowledged in the discussion.

-We appreciate this comment. In the GWAS, the measured serum vitamin B12 levels were quantile normalized based on ranking³⁰. The original study only provided that the median vitamin B12 level was 409 pmol/L (inter-quartile range: 305-555 pmol/L), suggesting a slightly skewed distribution of raw values. Due to insufficient data availability, we are not able to interpret our findings in the unit of pmol/L, though we fully agree that doing so would be more clinically relevant. We have discussed this limitation in Discussion:

“Fourth, our MR analyses could estimate population-averaged associations between vitamin B12 levels and the outcomes, but not potential dose-dependent effects of vitamin B12 levels. Furthermore, the distribution of raw vitamin B12 levels may be skewed, which may result in inconsistent interpretation of a one-standard-deviation change across the population. Investigating these aspects would require availability of both vitamin B12 measurements and the outcomes in the same study population^{37,38}.”

“We anticipate that future investigations will more comprehensively illuminate the role of vitamin B12 in various psychiatric disorders and cognitive impairment in diverse populations.”

Additional References

1. Howe LJ, Nivard MG, Morris TT, et al. Within-sibship genome-wide association analyses decrease bias in estimates of direct genetic effects. *Nat Genet.* 2022;54(5):581-592. doi:10.1038/s41588-022-01062-7
2. Brumpton B, Sanderson E, Heilbron K, et al. Avoiding dynastic, assortative mating, and population stratification biases in Mendelian randomization through within-family analyses. *Nat Commun.* 2020;11(1):3519. doi:10.1038/s41467-020-17117-4
3. Davies NM, Howe LJ, Brumpton B, Havdahl A, Evans DM, Davey Smith G. Within family Mendelian randomization studies. *Hum Mol Genet.* 2019;28(R2):R170-R179. doi:10.1093/hmg/ddz204
4. Bowden J, Davey Smith G, Haycock PC, Burgess S. Consistent Estimation in Mendelian Randomization with Some Invalid Instruments Using a Weighted Median Estimator. *Genet Epidemiol.* 2016;40(4):304-314. doi:10.1002/gepi.21965
5. Burgess S, Butterworth A, Thompson SG. Mendelian randomization analysis with multiple genetic variants using summarized data. *Genet Epidemiol.* 2013;37(7):658-665. doi:10.1002/gepi.21758
6. Backman JD, Li AH, Marcketta A, et al. Exome sequencing and analysis of 454,787 UK Biobank participants. *Nature.* 2021;599(7886):628-634. doi:10.1038/s41586-021-04103-z
7. Karczewski KJ, Gupta R, Kanai M, et al. Pan-UK Biobank GWAS improves discovery, analysis of genetic architecture, and resolution into ancestry-enriched effects. Published online October 1, 2024:2024.03.13.24303864. doi:10.1101/2024.03.13.24303864
8. Lee JJ, Wedow R, Okbay A, et al. Gene discovery and polygenic prediction from a genome-wide association study of educational attainment in 1.1 million individuals. *Nat Genet.* 2018;50(8):1112-1121. doi:10.1038/s41588-018-0147-3
9. Ghousaini M, Mountjoy E, Carmona M, et al. Open Targets Genetics: systematic identification of trait-associated genes using large-scale genetics and functional genomics. *Nucleic Acids Res.* 2021;49(D1):D1311-D1320. doi:10.1093/nar/gkaa840
10. Ochoa D, Hercules A, Carmona M, et al. Open Targets Platform: supporting systematic drug-target identification and prioritisation. *Nucleic Acids Res.* 2021;49(D1):D1302-D1310. doi:10.1093/nar/gkaa1027
11. Bycroft C, Freeman C, Petkova D, et al. The UK Biobank resource with deep phenotyping and genomic data. *Nature.* 2018;562(7726):203-209. doi:10.1038/s41586-018-0579-z

12. Kurki MI, Karjalainen J, Palta P, et al. Author Correction: FinnGen provides genetic insights from a well-phenotyped isolated population. *Nature*. 2023;615(7952):E19. doi:10.1038/s41586-023-05837-8
13. Buniello A, MacArthur JAL, Cerezo M, et al. The NHGRI-EBI GWAS Catalog of published genome-wide association studies, targeted arrays and summary statistics 2019. *Nucleic Acids Res*. 2019;47(D1):D1005-D1012. doi:10.1093/nar/gky1120
14. MacArthur J, Bowler E, Cerezo M, et al. The new NHGRI-EBI Catalog of published genome-wide association studies (GWAS Catalog). *Nucleic Acids Res*. 2017;45(D1):D896-D901. doi:10.1093/nar/gkw1133
15. Bohnsack BL, Hirschi KK. Nutrient regulation of cell cycle progression. *Annu Rev Nutr*. 2004;24:433-453. doi:10.1146/annurev.nutr.23.011702.073203
16. O'Leary F, Samman S. Vitamin B12 in health and disease. *Nutrients*. 2010;2(3):299-316. doi:10.3390/nu2030299
17. Stover PJ. Physiology of folate and vitamin B12 in health and disease. *Nutr Rev*. 2004;62(6 Pt 2):S3-12; discussion S13. doi:10.1111/j.1753-4887.2004.tb00070.x
18. Calderon-Ospina CA, Nava-Mesa MO. B Vitamins in the nervous system: Current knowledge of the biochemical modes of action and synergies of thiamine, pyridoxine, and cobalamin. *CNS Neurosci Ther*. 2020;26(1):5-13. doi:10.1111/cns.13207
19. Briani C, Dalla Torre C, Citton V, et al. Cobalamin deficiency: clinical picture and radiological findings. *Nutrients*. 2013;5(11):4521-4539. doi:10.3390/nu5114521
20. Green R, Allen LH, Bjørke-Monsen AL, et al. Vitamin B12 deficiency. *Nat Rev Dis Primer*. 2017;3(1):1-20.
21. Stabler SP. Vitamin B12 deficiency. *N Engl J Med*. 2013;368(2):149-160.
22. Lindenbaum J, Healton EB, Savage DG, et al. Neuropsychiatric disorders caused by cobalamin deficiency in the absence of anemia or macrocytosis. *N Engl J Med*. 1988;318(26):1720-1728. doi:10.1056/NEJM198806303182604
23. Roffman JL, Lamberti JS, Achtyes E, et al. Randomized multicenter investigation of folate plus vitamin B12 supplementation in schizophrenia. *JAMA Psychiatry*. 2013;70(5):481-489. doi:10.1001/jamapsychiatry.2013.900
24. Walker JG, Batterham PJ, Mackinnon AJ, et al. Oral folic acid and vitamin B-12 supplementation to prevent cognitive decline in community-dwelling older adults with depressive symptoms—the Beyond Ageing Project: a randomized controlled trial. *Am J Clin Nutr*. 2012;95(1):194-203.

25. Almeida OP, Ford AH, Flicker L. Systematic review and meta-analysis of randomized placebo-controlled trials of folate and vitamin B12 for depression. *Int Psychogeriatr.* 2015;27(5):727-737. doi:10.1017/S1041610215000046
26. Walker JG, Mackinnon AJ, Batterham P, et al. Mental health literacy, folic acid and vitamin B12, and physical activity for the prevention of depression in older adults: randomised controlled trial. *Br J Psychiatry.* 2010;197(1):45-54. doi:10.1192/bjp.bp.109.075291
27. Laird EJ, O'Halloran AM, Molloy AM, et al. Low vitamin B12 but not folate is associated with incident depressive symptoms in community-dwelling older adults: a 4-year longitudinal study. *Br J Nutr.* Published online 2021:1-8.
28. Dib MJ, Ahmadi KR, Zagkos L, et al. Associations of Genetically Predicted Vitamin B(12) Status across the Phenome. *Nutrients.* 2022;14(23). doi:10.3390/nu14235031
29. Hu Y, Yu M, Wang Y, et al. Exploring the Association between Serum B Vitamins, Homocysteine and Mental Disorders: Insights from Mendelian Randomization. *Nutrients.* 2024;16(13):1986. doi:10.3390/nu16131986
30. Grarup N, Sulem P, Sandholt CH, et al. Genetic architecture of vitamin B12 and folate levels uncovered applying deeply sequenced large datasets. *PLoS Genet.* 2013;9(6):e1003530. doi:10.1371/journal.pgen.1003530
31. Hartwig FP, Davey Smith G, Bowden J. Robust inference in summary data Mendelian randomization via the zero modal pleiotropy assumption. *Int J Epidemiol.* 2017;46(6):1985-1998. doi:10.1093/ije/dyx102
32. Bowden J, Davey Smith G, Burgess S. Mendelian randomization with invalid instruments: effect estimation and bias detection through Egger regression. *Int J Epidemiol.* 2015;44(2):512-525. doi:10.1093/ije/dyv080
33. Major Depressive Disorder Working Group of the Psychiatric Genomics Consortium. Electronic address: andrew.mcintosh@ed.ac.uk, Major Depressive Disorder Working Group of the Psychiatric Genomics Consortium. Trans-ancestry genome-wide study of depression identifies 697 associations implicating cell types and pharmacotherapies. *Cell.* 2025;188(3):640-652.e9. doi:10.1016/j.cell.2024.12.002
34. Burgess S, Davies NM, Thompson SG. Bias due to participant overlap in two-sample Mendelian randomization. *Genet Epidemiol.* 2016;40(7):597-608. doi:10.1002/gepi.21998
35. Friligkou E, Løkhammer S, Cabrera-Mendoza B, et al. Gene discovery and biological insights into anxiety disorders from a large-scale multi-ancestry genome-wide association study. *Nat Genet.* 2024;56(10):2036-2045. doi:10.1038/s41588-024-01908-2

36. Strom NI, Burton CL, Iyegbe C, et al. Genome-Wide Association Study of Obsessive-Compulsive Symptoms including 33,943 individuals from the general population. *Mol Psychiatry*. 2024;29(9):2714-2723. doi:10.1038/s41380-024-02489-6
37. Burgess S, Davies NM, Thompson SG, E. PIC-InterAct Consortium. Instrumental variable analysis with a nonlinear exposure-outcome relationship. *Epidemiology*. 2014;25(6):877-885. doi:10.1097/EDE.000000000000161
38. Lu T, Nakanishi T, Yoshiji S, Butler-Laporte G, Greenwood CMT, Richards JB. Dose-dependent Association of Alcohol Consumption With Obesity and Type 2 Diabetes: Mendelian Randomization Analyses. *J Clin Endocrinol Metab*. 2023;108(12):3320-3329. doi:10.1210/clinem/dgad324

Reviewers' comments:

Reviewer #1 (Remarks to the Author):

I commend the authors for their thorough clarification of this study. I have follow-up questions about instrument selection.

-We thank Reviewer 1 again for the helpful comments that improved our manuscript.

[Line 120] How many independent genetic variants had a p-value $< 2.2 \times 10^{-9}$ before restricting to exome-sequencing data? What is the r^2 for excluding correlated genetic variants? The analyses could have limited power because many non-coding variants were excluded. Also, Bonferroni-corrected threshold accounting for 22.9 million tested variants is not appropriate because not all the tested variants are independent. Overall, I think the authors applied a very stringent approach for instrument selection. Given this study reported null findings, I would recommend sensitivity analyses using a more lenient and common approach for instrument selection, i.e., independent ($r^2 < 0.001$) genetic variants with p value $< 5 \times 10^{-8}$ without restricting to coding variants.

-We thank Reviewer 1 for their thoughtful comments. We agree that the current instrument selection approach is relatively stringent. However, implementing a more standard approach as suggested requires access to the full, genome-wide summary statistics from the original GWAS¹. Unfortunately, the original study, published in 2013, did not provide the full summary statistics, which would have included non-coding variants from whole-genome sequencing data in the Icelandic cohort. We sent two requests to the authors but did not receive a response.

As a result, our analyses were based on the genome-wide significant loci reported in Table 1 of the original publication (attached below), where the original authors applied a Bonferroni correction accounting for 22.9 million tested variants (which we also acknowledge is overly conservative). Nonetheless, the original authors did not report any additional loci with a p-value between 5×10^{-8} and 2.2×10^{-9} .

The genetic instruments used in our study were the reported lead variants at each locus (Supplementary Table S1). Most of these variants are located on different chromosomes. For those that are on the same chromosome, they are at least 2 megabases apart and have an $r^2 < 0.001$ based on the UK Biobank European ancestry population.

We agree that applying a more lenient instrument selection approach could potentially increase the statistical power of MR analyses to detect associations. However, we posit that the magnitudes of such associations, if any, are likely to be limited, given that the current analyses are already relatively well-powered.

We have clarified in Methods and highlighted this limitation in Discussion:

“In total, 11 loci were significantly associated with serum vitamin B12 levels and two loci with folate levels (p-value <2.2x10⁻⁹; Bonferroni-corrected genome-wide significance threshold accounting for 22.9 million tested variants)¹. Lead variants at these loci with publicly available test statistics were used as genetic instruments.”

“Further research, such as larger GWAS for vitamin B12 levels with publicly available summary statistics, may help identify additional instruments for MR analyses, potentially allowing for a more lenient instrument selection approach and increasing the statistical power to detect potential associations.”

Table 1. Reported genomic loci associated with serum vitamin B12 levels from Grarup et al¹.

SNV name	Locus	Chr.	Position (build 36/hg18)	Annotation ¹	Alleles ² (effect/other)	EAF	Icelandic		Danish – Inter99		Danish – Health2006		Combined		I ² (P _{HET})
							Effect	P	Effect	P	Effect	P	N	P	
Novel loci															
rs2336573	CD320	19	8,273,709	G220R	T/C	0.031	0.32	1.1 × 10 ⁻⁵¹	0.22	0.0057	0.31	1.7 × 10 ⁻⁸	45,575	8.4 × 10 ⁻⁵⁹	41 (0.033)
rs1131603	TCN2	22	29,348,975	L376S	C/T	0.055	0.19	4.3 × 10 ⁻²⁸	0.33	1.8 × 10 ⁻⁹	0.33	5.3 × 10 ⁻¹⁷	45,575	4.9 × 10 ⁻⁴⁹	62 (0.0050)
rs3742801	ABCD4	14	73,828,759	E368K	T/C	0.294	0.045	5.3 × 10 ⁻⁸	0.093	7.6 × 10 ⁻⁴	0.083	4.5 × 10 ⁻⁵	45,571	1.7 × 10 ⁻¹³	0 (0.20)
rs2270655	MMAA	4	146,795,868	Q363H	G/C	0.941	0.066	3.5 × 10 ⁻⁵	0.30	2.8 × 10 ⁻⁷	0.25	5.8 × 10 ⁻⁸	45,576	2.2 × 10 ⁻¹³	79 (7.1 × 10 ⁻⁵)
rs12272669	MMACHC	1	45,747,242	R206Q	A/G	0.0022	0.51	3.0 × 10 ⁻⁹	-	-	-	-	-	-	-
Novel SNV associations in reported loci															
rs34324219	TCN1	11	59,379,954	D301Y	C/A	0.881	0.21	8.8 × 10 ⁻⁷¹	0.40	3.2 × 10 ⁻²³	0.30	3.5 × 10 ⁻²⁴	45,576	1.1 × 10 ⁻¹¹¹	70 (0.001)
rs7788053 ³	FUT6	19	5,783,209	P124S	A/G	0.254	0.046	2.1 × 10 ⁻⁷	0.050	0.076	0.070	0.00072	45,575	1.7 × 10 ⁻¹⁰	0 (0.64)
Reported associated SNVs															
rs602662	FUT2	19	53,898,797	G258S	A/G	0.596	0.16	4.1 × 10 ⁻⁹⁶	0.19	3.5 × 10 ⁻¹³	0.23	1.9 × 10 ⁻³⁴	45,568	2.4 × 10 ⁻¹³⁹	0 (0.14)
rs1801222	CUBN	10	17,196,157	F253S	G/A	0.593	0.11	1.1 × 10 ⁻⁵²	0.14	7.6 × 10 ⁻⁸	0.17	2.9 × 10 ⁻¹⁸	45,576	3.3 × 10 ⁻⁷⁵	0 (0.48)
rs41281112	CLYBL	13	99,316,635	R259X	C/T	0.948	0.17	9.6 × 10 ⁻²⁷	0.24	0.0013	0.29	2.5 × 10 ⁻⁷	45,576	8.9 × 10 ⁻³⁵	0 (0.90)
rs1141321 ⁴	MUT	6	49,520,392	R532H	C/T	0.627	0.061	1.4 × 10 ⁻¹⁶	0.12	1.4 × 10 ⁻⁵	0.11	1.4 × 10 ⁻⁷	45,574	3.6 × 10 ⁻²⁶	0 (0.24)

Association results for serum B₁₂ in Icelandic and Danish study samples separately and combined. The effect allele is the allele associated with increased serum B₁₂ levels. The effect is on a quantile normalized scale. Data were combined in fixed effect meta-analyses based on P-value and direction of effect adjusted for the number of individuals in each sample. Values of I² are percentages. Chr., chromosome; EAF, effect allele frequency; HET, heterogeneity; SNV, single nucleotide variant.
¹The annotation is based on the RefSeq hg18.
²The reference alleles based on Build 36 hg18 are shown in bold.
³In the Icelandic data the strongest signal at the FUT6 locus is for rs708686 located 5' to the FUT6 gene (see Table S3).
⁴Danish data are given for the perfect proxy rs4267943 (1000 Genomes data; r² = 1.0).
doi:10.1371/journal.pgen.1003530.t001

Reviewer #3 (Remarks to the Author):

I appreciate the authors' efforts to amend the manuscript, and do not have any further comments.

-We thank Reviewer 3 for reviewing our manuscript.

Reviewer #4 (Remarks to the Author):

I am happy with the author's responses and the corresponding revision which address most of my concerns.

-We thank Reviewer 4 for reviewing our manuscript.

Additional References

1. Grarup N, Sulem P, Sandholt CH, et al. Genetic architecture of vitamin B12 and folate levels uncovered applying deeply sequenced large datasets. *PLoS Genet.* 2013;9(6):e1003530. doi:10.1371/journal.pgen.1003530